# Heat Shock Factor 1-dependent extracellular matrix remodeling mediates the transition from chronic intestinal inflammation to colon cancer

Oshrat Levi-Galibov[1], Hagar Lavon[1], Rina Wassermann-Dozorets[1], Meirav Pevsner-Fischer[1], Shimrit Mayer[1], Esther Wershof[2], Yaniv Stein[1], Lauren E. Brown [3], Wenhan Zhang[3], Gil Friedman[1], Reinat Nevo[1], Ofra Golani [4], Lior H. Katz[5,6], Rona Yaeger[7], Ido Laish[5,8], John A. Porco [3], Erik Sahai [2], Dror S. Shouval[8,9], David Kelsen [7] & Ruth Scherz-Shouval [1✉]

In the colon, long-term exposure to chronic inflammation drives colitis-associated colon cancer (CAC) in patients with inflammatory bowel disease. While the causal and clinical links are well established, molecular understanding of how chronic inflammation leads to the development of colon cancer is lacking. Here we deconstruct the evolving microenvironment of CAC by measuring proteomic changes and extracellular matrix (ECM) organization over time in a mouse model of CAC. We detect early changes in ECM structure and composition, and report a crucial role for the transcriptional regulator heat shock factor 1 (HSF1) in orchestrating these events. Loss of HSF1 abrogates ECM assembly by colon fibroblasts in cell-culture, prevents inflammation-induced ECM remodeling in mice and inhibits progression to CAC. Establishing relevance to human disease, we find high activation of stromal HSF1 in CAC patients, and detect the HSF1-dependent proteomic ECM signature in human colorectal cancer. Thus, HSF1-dependent ECM remodeling plays a crucial role in mediating inflammation-driven colon cancer.

[1] Department of Biomolecular Sciences, The Weizmann Institute of Science, Rehovot, Israel. [2] The Francis Crick Institute, London, UK. [3] Department of Chemistry and Center for Molecular Discovery (BU-CMD), Boston University, Boston, MA, USA. [4] Department of Life Sciences Core Facilities, The Weizmann Institute of Science, Rehovot, Israel. [5] Gastroenterology Institute, Sheba Medical Center, Tel Hashomer, Ramat Gan, Israel. [6] Department of Gastroenterology and Hepatology, Hadassah Medical Center, Jerusalem, Israel. [7] Gastrointestinal Oncology Service, Memorial Sloan Kettering Cancer Center, and Weil Cornell Medical College, New York, NY, USA. [8] Sackler Faculty of Medicine, Tel-Aviv University, Tel Aviv, Israel. [9] Pediatric Gastroenterology Unit, Edmond and Lily Safra Children's Hospital, Sheba Medical Center, Ramat Gan, Israel. ✉email: ruth.shouval@weizmann.ac.il

Colorectal cancer (CRC) is one of the leading causes of cancer-related deaths, world-wide. CRC is usually caused by a combination of genetic and environmental factors, leading to dysplastic lesions and eventually carcinoma. A particularly aggressive subtype of CRC is colitis-associated colon cancer (CAC), arising in patients with inflammatory bowel disease (IBD) due to long-term exposure to chronic inflammation[1]. Different than the adenoma-dysplasia-cancer progression of sporadic CRC, CAC develops from an accumulation of somatic mutations that mediate progression from inflamed mucosa to dysplasia to carcinoma[2,3]. The spectrum of genomic alterations in CAC is distinct from that of sporadic CRC—alterations in TP53, IDH1, and MYC are significantly more frequent in CAC, and mutations in APC are significantly less frequent, than those reported in sporadic CRC[2]. Both the initiation and the progression of CAC are expedited by the inflammatory insult[4,5]. Such inflammatory signals are mediated through dynamic crosstalk between cancer cells and cells of the tumor microenvironment (TME). The TME comprises various types of non-malignant stromal cells, including macrophages, neutrophils, lymphocytes, endothelial cells, and cancer-associated fibroblasts (CAFs).

Fibroblasts are major constituents of the normal colon, where they are mostly quiescent and function to maintain ECM integrity and limit epithelial cell proliferation and differentiation[6]. Fibroblasts provide fibers and connector proteins that give structure to tissue, and produce specialized ECMs by expressing and secreting a variable repertoire of structural proteins[7]. In severely injured or chronically inflamed tissues, fibroblasts produce excessive ECM, which, without reciprocally balanced degradation, results in fibrosis and can eventually lead to cancer[7–9]. Co-evolving with the cancer cells, fibroblasts in tumors are recruited and rewired to become protumorigenic CAFs that support critical aspects of malignancy by secretion of cytokines, chemokines, and growth factors[10]. CAFs secrete distinct matrix components and remodeling enzymes from those secreted by normal fibroblasts, resulting in aberrant ECM composition and topography which contribute to increased stiffness of the tumor[11] and promote tumor invasion and progression[12]. Moreover, proteases secreted by CAFs cleave and activate growth factors, cytokines, and cell adhesion molecules (CAMs) embedded in the ECM, increasing cancer cell motility and EMT[6]. Unlike cancer cells, CAFs are genomically stable, and do not harbor oncogenic mutations that could drive their co-evolution and functional reprogramming. Instead, stromal reprogramming is achieved by massive transcriptional rewiring[13–18].

Heat shock factor 1 (HSF1), master regulator of the heat-shock response, enables the transcriptional rewiring of fibroblasts into protumorigenic CAFs[13]. HSF1 has historically been studied in the context of thermal stress. A key role in cancer was established when Hsf1 null mice were found to be extremely resistant to tumorigenesis[19]. This phenomenon was initially attributed mainly to HSF1's activity in cancer cells[19,20]; however, a complementary crucial role for HSF1 was recently demonstrated in CAFs[13,21]. The stromal HSF1 program is completely different from the program HSF1 drives in the adjacent cancer cells[13,20] or during heat-shock. In fibroblasts co-cultured with cancer cells HSF1 drives the expression of genes involved in adhesion and wound healing, leading to activation of genes involved in ECM organization in adjacent cancer cells[13,21]. This is a fundamental survival mechanism that appears to have been subverted to support the growing tumor in a non-cell-autonomous manner.

As a pro-survival factor, HSF1 was previously shown to play a protective role in acute colitis in mice[22]. When exposed to one course of dextran sulfate sodium (DSS) treatment (7 days), Hsf1 null mice displayed a more severe form of colitis compared to their wild-type (WT) counterparts. The effect of HSF1 was mediated through heat shock protein (HSP)70, and involved suppression of proinflammatory cytokines and CAMs[22]. In colitis-induced cancer, however, HSF1's protective activity was recently shown to promote cancer[23]. Chronic exposure to DSS in a model of azoxymethane (AOM)–DSS colitis-induced cancer led to HSF1-dependent activation of mTOR and increased glutaminolysis in cancer cells, and promoted tumor growth.

In this work, we ask whether the protumorigenic role of HSF1 in colitis-induced cancer includes non-cell-autonomous effects on the TME. We find that HSF1 is activated in stromal fibroblasts during early stages of inflammation, and its activation leads to ECM remodeling, supporting the development of colon cancer. Using the AOM–DSS model we analyze the ECM at different time points and find that both the structure and the composition of the ECM change long before tumors are observed. These inflammation-induced changes are HSF1-dependent, as is the consequent progression to CAC. In patients, we find high activation of stromal HSF1 in CAC, and high conservation of the HSF1-dependent proteomic ECM signature in human CRC, confirming the relevance of our findings to human disease and highlighting the crucial role of stromal HSF1 in CAC.

## Results

**Loss of HSF1 abrogates ECM secretion by fibroblasts**. In a variety of human carcinomas, HSF1 is activated in CAFs. When co-cultured with cancer cells, fibroblasts express an HSF1-dependent transcriptional program, which includes genes involved in wound-healing and ECM remodeling[13]. To directly assess the effect of stromal HSF1 on cancer-dependent ECM assembly, we compared the ability of WT and Hsf1 null mouse embryonic fibroblasts (MEFs) to deposit fibrillar collagen[24], in vitro, in the presence of conditioned media from two different cancer cell-lines, using second harmonic generation (SHG). Loss of Hsf1 in MEFs impeded ECM deposition induced by MC38 colon cancer-conditioned media (Fig. 1a, b) and 4T1 breast cancer-conditioned media (Fig. 1c, d and Supplementary Fig. 1). The surface area covered by secreted collagen was reduced in Hsf1 null MEFs compared to WT counterparts (Fig. 1a–d), suggesting that stromal HSF1 is required for proper ECM assembly by cancer-conditioned fibroblasts.

**Loss of HSF1 attenuates inflammation-induced colon cancer**. To test whether HSF1-dependent ECM remodeling promotes cancer in vivo, we exposed WT and Hsf1 null mice to an inflammation-driven colon cancer model[25]. In this model, mice are injected with a carcinogen, AOM, followed by two cycles of treatment with an inflammation-inducing irritant, DSS (1.5%, 5 days), and sacrificed at day 52 post AOM injection (Fig. 2a). Both WT and Hsf1 null mice experienced transient weight loss following each course of DSS, with no significant differences between the two genotypes (Fig. 2b). A similar effect was observed following acute DSS treatment (without AOM; one cycle of 1.5% DSS, 7 days; Supplementary Fig. 2a, b). Nevertheless, assessment of the disease activity index (DAI; see the "Methods" section) indicated that WT mice were sicker than Hsf1 null mice (Fig. 2c, Supplementary Fig. 2c, Supplementary Table 1). The DAI of WT mice increased significantly more than that of Hsf1 null mice following each cycle of DSS in the AOM-DSS protocol (day 11–20 and day 32–52, Fig. 2c), and the WT mice did not recover from the second cycle of DSS. While a similar trend was observed in response to acute DSS treatment, this trend was not significant (Supplementary Fig. 2c).

To better assess these effects we performed colonoscopy on the AOM-DSS treated mice at day 52, and then sacrificed the mice and examined their colons. Colonoscopy and post-mortem

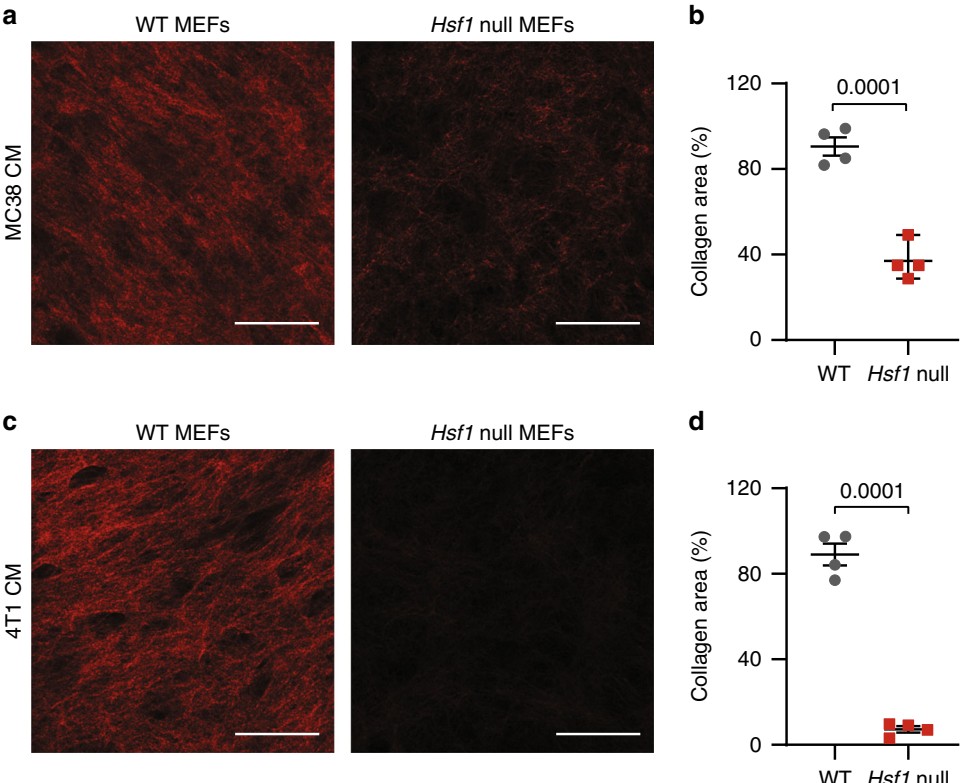

**Fig. 1 Loss of HSF1 impedes fibrillar collagen deposition by fibroblasts.** WT or *Hsf1* null immortalized MEFs were induced to secrete ECM by 7 days incubation with conditioned media from MC38 colon cancer cells **a**, **b** or 4T1 breast cancer cells **c**, **d** supplemented by growth factors and insulin. Representative images are shown in **a**, **c**. The average area covered by collagen in *n* = 4 biological independent cells, examined over four independent experiments, each with three technical replicates, is quantified in **b**, **d**. Scale bar—50 μm. Results are shown as mean ± SEM. *P* value was calculated using a two-sided unpaired Student's *t*-test.

macroscopic examination of the colons revealed significant differences between WT and *Hsf1* null genotypes: not only was tumor burden significantly lower in *Hsf1* null mice compared to WT (Fig. 2d, e, Supplementary Table 1), but *Hsf1* null mice also exhibited significantly reduced inflammation at this time point (Fig. 2f, g, Supplementary Table 1), suggesting that the observed effect of HSF1 on tumor burden may be mediated through stromal effects on inflammation.

**Loss of HSF1 attenuates fibrosis.** Inflammation is well known to lead to fibrosis. To test whether HSF1 affects fibrosis in the AOM-DSS model, we examined collagen deposition at day 52 by Masson trichrome and Sirius red staining. Both stains revealed increased collagen-rich fibrotic regions in WT colons compared to colons from *Hsf1* null mice, suggesting that the ECM is affected by this treatment in an HSF1-dependent manner (Fig. 3a). To better assess these effects, we performed SHG imaging of the affected colon crypts, as well as normal crypts from sham-treated mice (Fig. 3b). While the structure and amount of fibrillar collagen secreted in colon crypts from sham-treated animals was similar between WT and *Hsf1* null mice (Supplementary Fig. 3), we observed stark differences between the ECM of AOM-DSS-treated WT vs. *Hsf1* null mice (Fig. 3b–l): WT mice treated with AOM-DSS exhibited extensive fibrosis and increased thickening of the inter-crypt walls, leading to reduction in the number of colon crypts (Fig. 3c), increase in the size of the crypts (Fig. 3e), and increase in the distance between crypts (Fig. 3g, h, j, k) as compared to *Hsf1* null mice, which exhibited normal ECM morphology. The distorted ECM characteristics significantly correlated with tumor burden, as mice with higher tumor burden

had fewer and larger crypts (Fig. 3d, f), with increased distance between crypts (Fig. 3i, l) than mice with lower tumor burden.

**HSF1-dependent inflammation-induced ECM remodeling precedes tumor formation.** The aberrant ECM morphology found following the AOM-DSS treatment could be a result of the heavy tumor burden on the tissue. Alternatively, HSF1-dependent fibrosis and ECM remodeling could be driving the transition from inflammation to cancer, in which case we would expect changes in the ECM to precede the appearance of tumors. To investigate this, we sacrificed AOM-DSS-treated mice at two earlier time points, following one cycle of DSS—15 and 20 days post AOM injection (Supplementary Fig. 4a). Acute colitis symptoms usually peak at day 15 and decrease at day 20, as observed by the transient and concomitant changes in body weight and DAI (Supplementary Fig. 4b, c). Importantly, tumors were not observed by macroscopic or histopathological evaluation at these timepoints (Supplementary Fig. 4d and Supplementary Table 1). Nevertheless, colonoscopy examination revealed significant inflammation in WT mice at day 20, but not in *Hsf1* null mice (Fig. 4a), suggesting that HSF1 is required for the early inflammatory response that leads to cancer.

Histopathological examination of colon tissue revealed that even in this early, pre-malignant stage, WT mice exhibit significantly more inflammation and crypt damage than *Hsf1* null mice (Fig. 4b, c and Supplementary Table 1). Sirius red and Masson trichrome staining confirmed that colons from WT mice treated with AOM–DSS exhibited more collagen-rich fibrotic regions than colons from *Hsf1* null mice (Fig. 4d, e). SHG imaging of the colon crypts showed that the ECM of WT mice at day 20 was highly

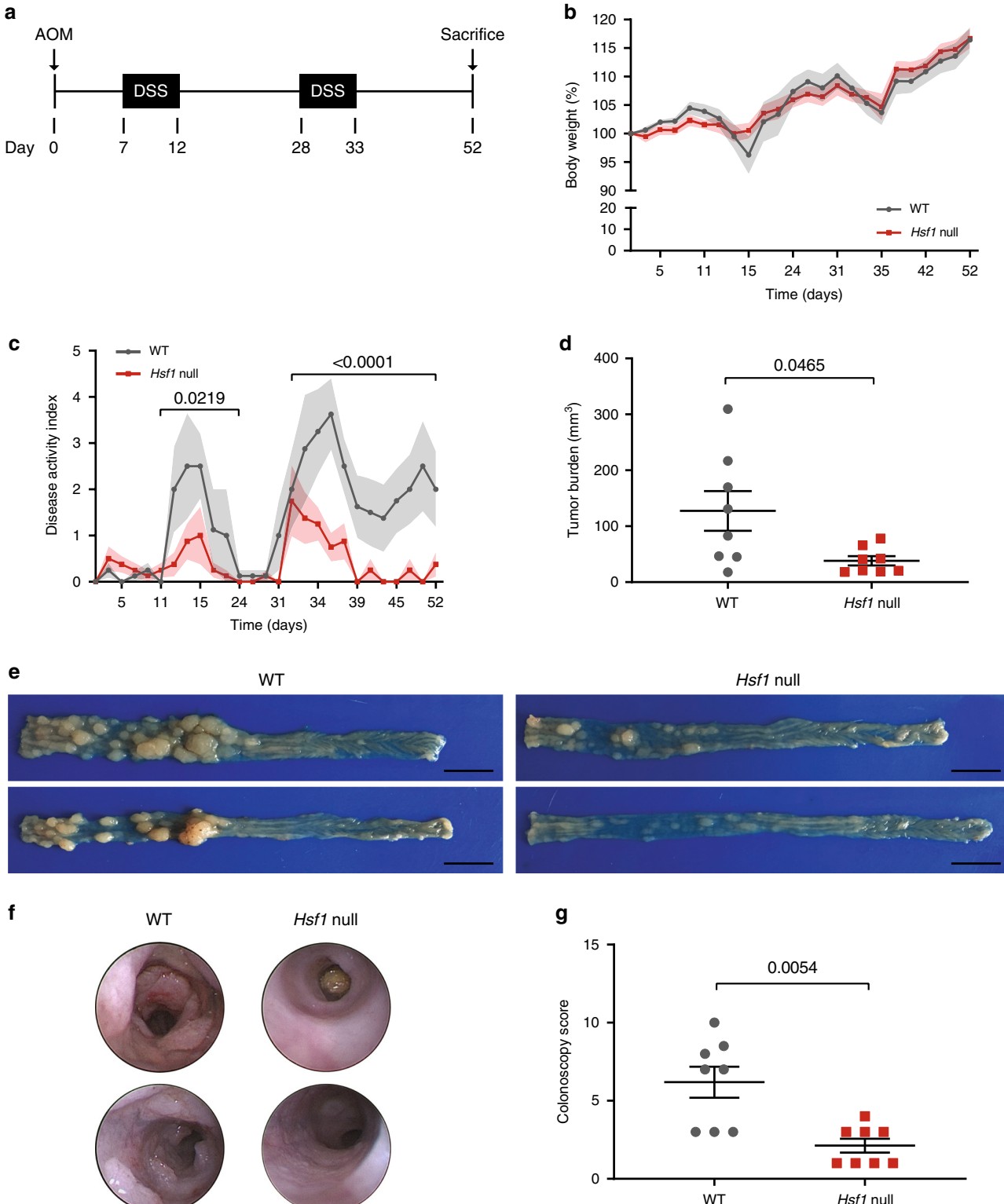

**Fig. 2 *Hsf1* null mice are highly resistant to inflammation-induced cancer. a** Schematic representation of the experimental protocol. WT and *Hsf1* null mice (*n* = 8 per group, combined from two independent experiments) were injected intraperitoneally with AOM (10 mg/kg), followed by 2 × 5-day cycles of 1.5% DSS in the drinking water. **b** The changes in body weight of the mice throughout the experiment, and **c** assessment of the disease activity index (DAI) are plotted and presented as mean ± SEM, analyzed by two-way ANOVA (with discrete time point and genotype as the independent variables) utilizing Tukey's test to account for multiple comparisons. **d, e** Tumor burden calculation based on macroscopic measures of tumors at day 52 of the AOM-DSS protocol. Representative images of two WT (left) and two *Hsf1* null (right) mouse colons are shown in **e**. Scale bar—1 cm. **f, g** Prior to sacrificing the mice at day 52 of the AOM-DSS protocol, colonoscopies were performed and inflammation was scored as described in the "Methods" section. Representative images from 2 WT and 2 *Hsf1* null mice are shown in **f**. Colonoscopy scores are shown in **g**. Results for **d, g** are shown as mean ± SEM, *p* value was analyzed using a two-sided Welch's test for **d** and by a nonparametric Mann–Whitney test for **g**.

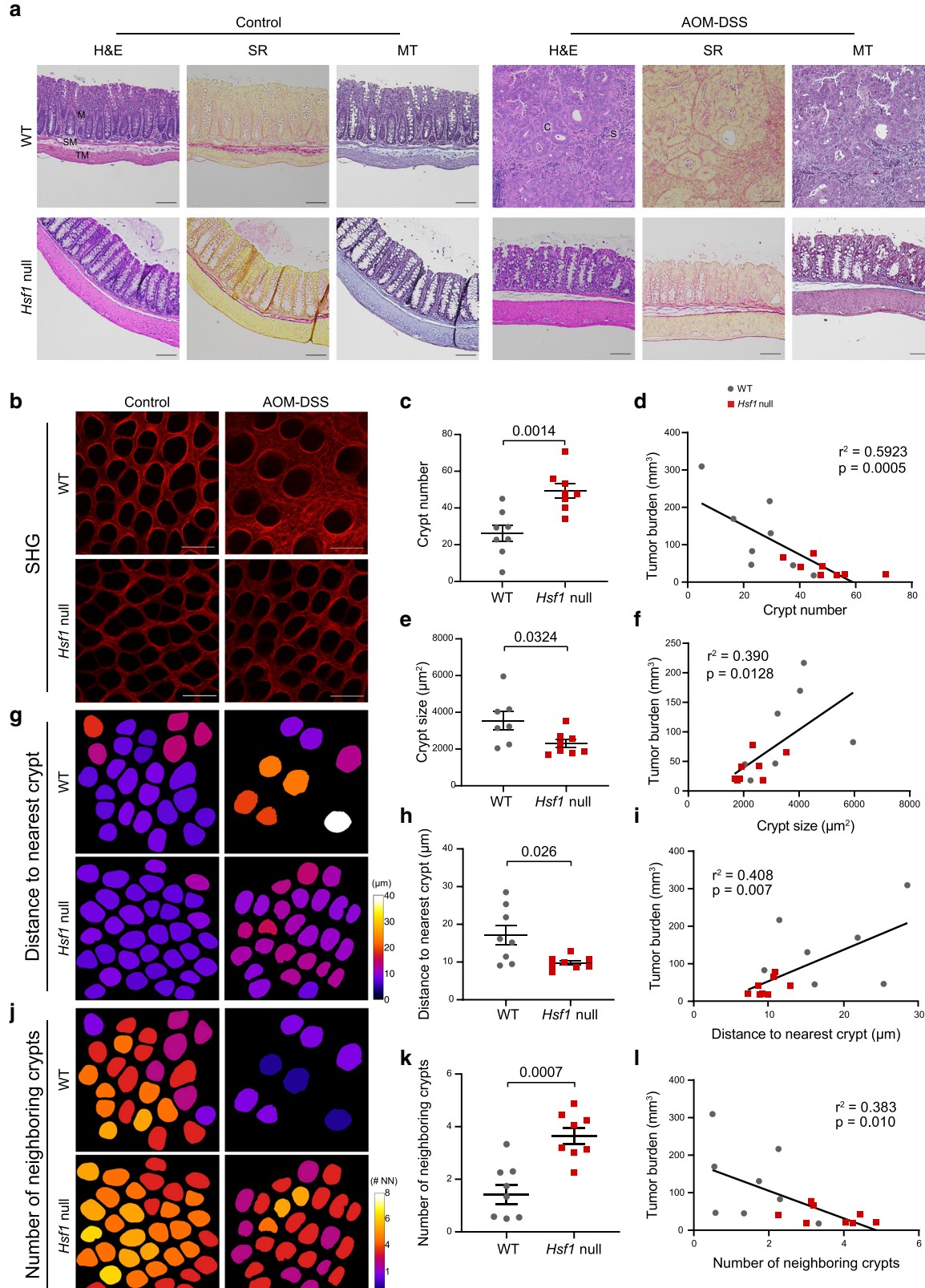

distorted, while colons from *Hsf1* null mice had a normal appearance, comparable to that of control mice (Fig. 4f–j and Supplementary Fig. 3b–e). Colon crypts of WT mice were larger, fewer, and more distant apart than *Hsf1* null colon crypts (Fig. 4g–j). Examination of *Hsf1* null mice at the earlier timepoint of 15 days revealed that they did in fact exhibit some ECM rearrangements

(Fig. 4f), accompanied by a slight reduction in crypt number (compared to day 20, Fig. 4g and Supplementary Fig. 3b), and increase in crypt damage (Supplementary Table 1) however these changes were not significant and were resolved by day 20.

Immune profiling of colons from day 15 treated WT mice revealed a significant increase in the number of neutrophils

**Fig. 3 *Hsf1* null mice exhibit reduced ECM rearrangements in colon cancer. a** Representative H&E, Sirius red (SR), and Masson's trichrome (MT) staining of colons from WT (upper panels) and *Hsf1* null (lower panels) mice following 52 days treatment with AOM-DSS (right panels) or control (sham injection and water; left panels). Scale bar—100 μm; M—mucosa; SM—submucosa; TM—tunica muscularis; C—cancer; S—stroma. **b–l** Representative cross-sections of the colons with second harmonic generation (SHG) images taken from the mucosal side **b** and analysis **c–l** of fibrillar collagen of mouse colons following 52 days of AOM-DSS treatment or control. Scale bar—100 μm. **c–f** Quantification of the average crypt number **c** and size **e**, and Pearson correlation (two-sided) between tumor burden and crypt number **d** and size. **g–h** Analysis of the average distance to the nearest crypt, within 40 μm, calculated based on SHG images of fibrillar collagen of mouse colons following 52 days of AOM-DSS treatment or control. Representative distance heatmaps of the SHG images presented in **b** are shown in **g**. **i** Pearson correlation between the average distance to the nearest crypt and tumor burden. **j–k** Analysis of the number of neighboring crypts within 20 μm, calculated based on SHG images of fibrillar collagen of mouse colons following 52 days of AOM-DSS treatment or control. Representative distance heatmaps of the SHG images presented in **b** are shown in **k**. #NN—number of nearest neighbors. **l** Pearson correlation between the number of neighboring crypts and tumor burden. $n = 8$ mice combined from two independent experiments (one outlier was removed in **e**, see the "Methods" section for details). **c**, **e**, **h**, **k** results are shown as mean ± SEM, **h** $p$ value was analyzed using two-sided Welch's $t$-test, **e**, **h**, and **k** were analyzed by an unpaired Student's $t$-test.

(CD45$^+$CD11b$^+$Ly6G$^+$), macrophages (CD45$^+$CD11b$^+$Ly6G$^-$F4/80$^+$), and monocytes (CD45$^+$CD11b$^+$Ly6G$^-$F4/80$^-$Ly6C$^+$) in the epithelial fraction (EF), as compared to colons from sham-treated mice (Supplementary Fig. 4e–h). *Hsf1* null mice showed an intermediate phenotype: the above-mentioned cell populations were somewhat increased compared to sham-treated colons, however the differences were not statistically significant. The increase in myeloid immune cells was transient, both in WT and in *Hsf1* null mice, as these populations returned to control levels at day 20 of AOM–DSS treatment (Supplementary Fig. 4i–k). These results suggest that HSF1-induced ECM remodeling precedes tumor formation, and is required for inflammation-driven colon cancer. They further suggest that HSF1 is not required for initiation of acute inflammation, yet it is required for the ensuing response to chronic inflammation.

Supporting this notion, histopathological examination of colons from mice treated with the acute DSS protocol (DSS only, without AOM; 8 days post treatment) revealed a modest and similar rise in the inflammation score of WT and *Hsf1* null mice (Supplementary Fig. 5a, b and Supplementary Table 1). SHG imaging also showed moderate and non-significant changes in crypt number, size, and distance that were comparable between WT and *Hsf1* null mice (Supplementary Fig. 5c–i).

Taken together, these experiments suggest that an acute and transient inflammatory signal leads to minor and transient changes to the ECM which are HSF1-independent.

Together with an oncogenic driver, and in the context of chronic inflammation, HSF1 is activated, leading to massive ECM remodeling and, consequently, inflammation-driven colon cancer.

**Loss of *Hsf1* affects ECM protein composition.** To better understand the mechanism by which HSF1 affects ECM remodeling, we performed mass spectrometry analysis of colons from WT and *Hsf1* null mice at the pre-malignant stage (day 20) compared to control mice (day 0). We also analyzed colons at the late, malignant stage (day 52; see the "Methods" section for details). 3830 proteins were detected in samples from days 0–20, and 4234 proteins were detected at day 52 (Supplementary Data 1). Of these, only 11 proteins were differentially expressed between WT and *Hsf1* null mice at day 0 (Fig. 5a and Supplementary Data 2). 20 days of treatment with AOM-DSS led to massive changes in protein expression in WT colons, and to a lesser extent in *Hsf1* null colons (Fig. 5b and Supplementary Data 3). At this pre-malignant stage, 42 proteins were differentially upregulated in WT compared to *Hsf1* null colons and 18 were downregulated (Fig. 5a and Supplementary Data 2). Approximately a third of these differentially expressed proteins were also differentially expressed at day 52, where a total of 136 proteins were upregulated and 178 were downregulated in

WT compared to *Hsf1* null colons (Fig. 5a and Supplementary Data 2).

To define potential pathways affected by HSF1 in our model we performed pathway analysis on the top differentially expressed proteins (Fig. 5b, Supplementary Fig. 6a, and Supplementary Data 3; see the "Methods" section). Following 20 days of AOM-DSS treatment, the most significantly upregulated proteins in WT compared to *Hsf1* null colons were ECM proteins (FN1, LAMA1), proteins involved in inflammation, wound healing, and innate immune responses (NFκB2, LCN2, S100A8), and proteins involved in complement and coagulation (C3, TF; Fig. 5b; Cluster 6, and Supplementary Data 4). Proteins involved in oxidation–reduction responses and metabolic pathways (ACAD10, ALDH1B1) were downregulated in 20-day treated colons, and more so in WT colons compared to *Hsf1* null colons (Fig. 5b; Clusters 2 and 5, and Supplementary Data 4). Most of the differentially expressed proteins exhibited a similar trend of expression at day 52 of the AOM-DSS treatment (Supplementary Fig. 6b, c and Supplementary Data 4).

Since HSF1 is a transcriptional regulator, we performed RNA-sequencing of colons from the pre-malignant stage to assess whether the observed changes to the proteome are reflected in the transcriptome. Changes to the transcriptome are expected to precede changes to the proteome, and therefore we collected colons from day 15 AOM–DSS-treated mice, and compared their RNA expression profile to that of mice from day 0 (Supplementary Data 5). Hierarchical clustering highlighted 3506 differentially expressed genes between day 0 and day 15 WT and *Hsf1* null mice (Supplementary Fig. 6e and Supplementary Data 6). The most differentially upregulated genes in AOM–DSS-treated WT mice compared to *Hsf1* null mice were ECM constituents (*Col18a1, Fbln1, Lama1, Bgn*), genes involved in cytokine binding (*Csf1R, Il1r1, Tgfbr2*), and genes involved in inflammatory responses (*S100a8, Lcn2, Cxcl12*; Cluster 3, Supplementary Fig. 6e and Supplementary Data 7). Genes involved in oxidation–reduction and metabolic pathways (*Acad12, Aldh2, Cox7a1*) were most differentially downregulated in 15 day-treated WT mice, compared to *Hsf1* null mice (Cluster 1, Supplementary Fig. 6e and Supplementary Data 7). Together with the proteomic changes described above, these findings support the notion that HSF1 drives the expression of genes involved in ECM remodeling and inflammatory responses, leading to the proteomic changes observed at day 20.

To further dissect the changes in ECM-associated proteins we focused on matrisome proteins using the ECM protein expression database MatrisomeDB[26,27]. This database contains ~1000 core matrisome and matrisome-associated proteins, 107 of which were previously identified in the normal murine colon[28]. 188 matrisome proteins were detected in our mass-spec analysis of day 0–20, 82 of which overlapped with the colon matrisome (Fig. 5c and Supplementary Data 8). 193 matrisome proteins were

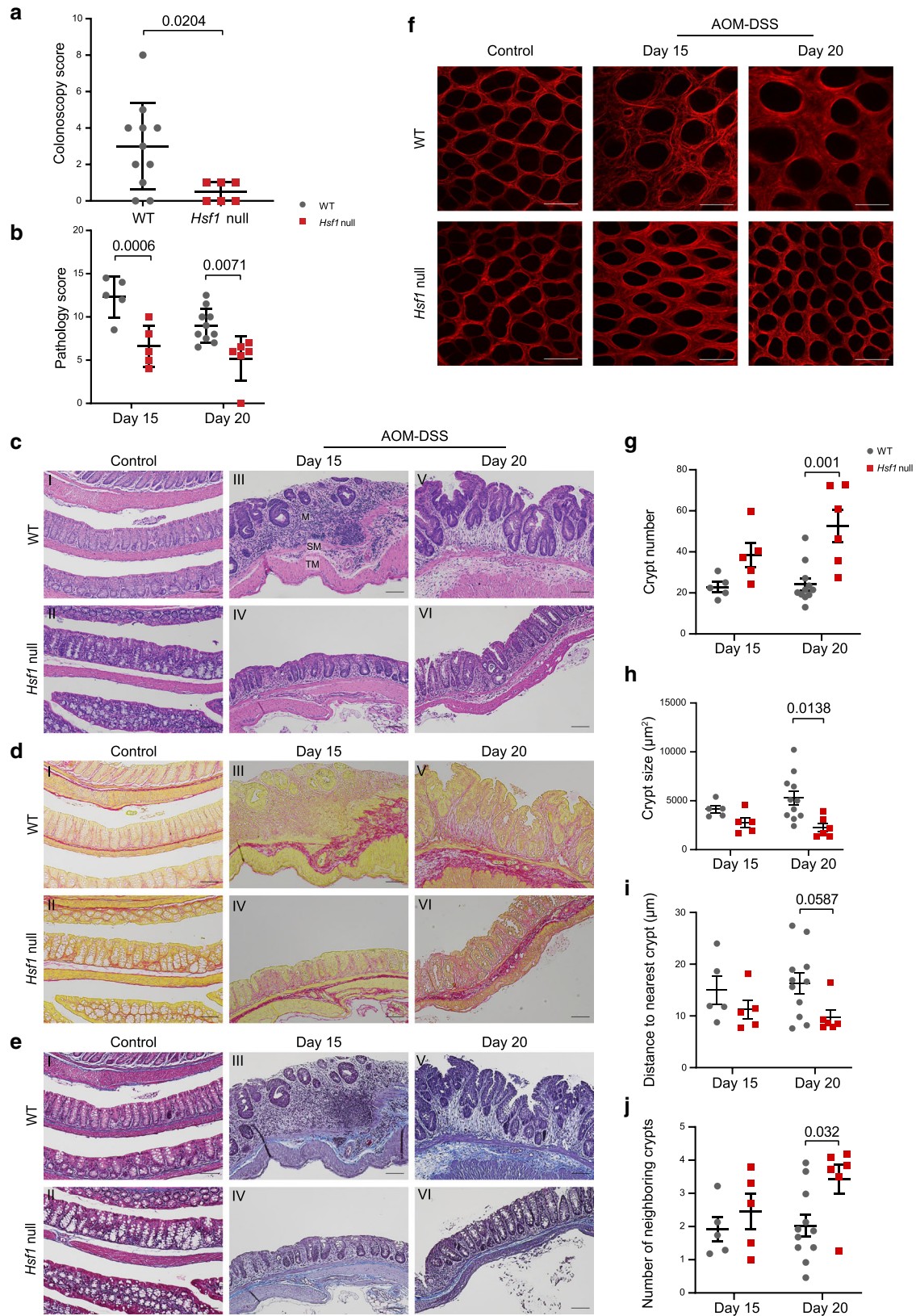

detected at day 52, 80 of which overlapped with the colon matrisome (Fig. 5c and Supplementary Data 8). These numbers suggest that we detected the majority of known colon matrisome proteins (~75%), as well as many matrix proteins which have not been previously associated with the mouse colon. Next, we asked which of these matrix proteins is differentially expressed in our

mass spectrometry dataset. We found 39 core matrisome and matrisome-associated proteins differentially expressed between control and 20-day treated WT and *Hsf1* null colons (Fig. 5d, e). Of these, 17 were previously identified as part of the murine colon matrisome, and 22 proteins, including ECM regulators (such as SERPINs) and secreted matrisome-associated proteins (such as

**Fig. 4 HSF1-dependent ECM remodeling precedes tumor growth.** WT and *Hsf1* null mice were injected intraperitoneally with AOM (10 mg/kg), followed by 5 days of 1.5% DSS in the drinking water, and sacrificed 3 or 8 days later (day 15 and 20, respectively). Age matched non-treated mice were used as control and sacrificed at day 0. **a** Prior to sacrificing the mice at day 20 of the AOM-DSS protocol, colonoscopies were performed and inflammation was scored, presented as mean ± SEM and analyzed by a nonparametric Mann–Whitney test (two-sided). **b**, **c** Colon sections were fixed, stained with H&E and inflammation was scored by a pathologist. Representative H&E images are shown in **c**. M—mucosa; SM—submucosa; TM—tunica muscularis. **d**, **e** Sirius red (SR; **d**) and Masson's trichrome (MT; **e**) staining of colons from WT (upper panels) and *Hsf1* null (lower panels) mice following 15 and 20 days treatment with AOM-DSS (right panels) or control (left panel). Scale bar—100 μm. **f** Representative second harmonic generation (SHG) images of fibrillar collagen of mouse colons following 15 and 20 days of AOM-DSS treatment or control. Scale bar—100 μm. **g–j** Quantification of the average crypt number **g**; crypt size **h**; distance to the nearest crypt (within 40 μm) **i**; and number of neighboring crypts (within 20 μm) **j** in mouse colons following 15 and 20 days of AOM-DSS treatment based on SHG signal. **b**, **g–j** results presented as mean ± SEM, analyzed by two-way ANOVA and Bonferroni correction for multiple comparisons. $n = 5$ mice for both genotypes at day 15, examined over two independent experiments; $n = 11$ WT mice and $n = 6$ *Hsf1* null mice for day 20 AOM-DSS treated mice, $n = 2$ mice for both genotypes for control; combined from two independent experiments.

S100A8 and S100A9), were not (Supplementary Data 8). 19 core matrisome and matrisome-associated proteins were differentially expressed between WT and *Hsf1* null colons at day 52 of the AOM-DSS treatment, only 10 of which were previously identified as part of the murine colon matrisome (Fig. 5d and Supplementary Fig. 5d).

Gene expression analysis of our RNA-seq dataset revealed 234 core matrisome and matrisome-associated genes differentially expressed between colons from day 0 and day 15 WT and *Hsf1* null mice, the majority of which were upregulated in day 15 WT mice compared to all other groups (Supplementary Fig. 6f and Supplementary Data 9). Gene products of 41 of these were previously identified as part of the murine colon matrisome (Supplementary Data 9).

The fact that ~75% of the known colon matrisome was detected in our proteomic analysis (Fig. 5c), and yet ~50% of the differentially expressed proteins in our analysis were not previously identified in the colon matrisome (Fig. 5d), suggests that these proteins are specifically upregulated in inflammation and cancer and not in the normal colon. Moreover, these findings highlight HSF1 as a major regulator of the matrisome in inflammation-induced colon cancer.

**MMPs are upregulated and co-expressed with HSF1 in inflamed stromal cells.** Since mass-spec analysis takes a crude average of all cells and ECM in the colon, we next validated our mass-spec results by multiplexed immunofluorescent (MxIF) staining with antibodies targeting HSF1 and a selected list of potential targets. We chose for this follow-up analysis the core matrix protein FN1[28], the matrix remodeling enzymes MMP7[29] and MMP9[30], the collagen-specific chaperone SERPINH1[31] (also known as HSP47), and the neutrophil-associated proteins LCN2[32] and S100A8[33,34].

To achieve a comprehensive understanding of the dynamic changes that occur during inflammation and tumor progression we stained colon sections from day 0, 15, 20, and 52 of the AOM-DSS protocol, and scored expression levels and colocalization of selected targets (see the "Methods" section for details). αSMA was used as a marker of activated fibroblasts[35] (Fig. 6a). We also scored HSF1 nuclear expression as a proxy for its activation[13,36], and found that its expression in the stroma (Supplementary Fig. 7a), but not in the epithelial cells of the crypt (Supplementary Fig. 7b), reflected the disease activity (Fig. 2c) measured at the different timepoints.

At day 0, HSF1 was scarcely detected in the stroma. At day 15 of AOM-DSS—the day at which inflammation peaks and massive stromal infiltration is observed—HSF1 was strongly activated, mostly in αSMA-positive fibroblasts (Fig. 6a and Supplementary Fig. 7c, d). Similar to the DAI, activation of stromal HSF1 subsided at day 20 and increased again at day 52 (Supplementary Fig. 7a). Epithelial HSF1 showed a different

pattern of activation—it was active in normal epithelial cells of the crypts at day 0, its activity gradually declined in the inflamed crypts at days 15 and 20, and increased again at day 52, in the transformed epithelial cells of the tumors (Supplementary Fig. 7b).

MMP7, MMP9, and LCN2 all showed HSF1-dependent upregulation in the stroma in response to the AOM-DSS treatment (Fig. 6a, d). While these proteins were expressed to some extent also in the colon crypts, there was no clear trend suggesting inflammation-dependent or HSF1-dependent expression in epithelial cells of the crypts (Supplementary Fig. 7f–h). Next we asked whether these potential HSF1 targets are co-expressed with HSF1 in stromal cells. Indeed, we found that MMP7 and MMP9 partially overlapped with stromal HSF1 (Fig. 6e–g). LCN2 did not overlap with HSF1 however it did overlap with MMP9, most likely in neutrophils (Fig. 6a and Supplementary Fig. 6i).

S100A8, which together with S100A9 forms the calprotectin heterocomplex, serves as a biomarker for IBD[33,34]. Immunostaining shows that S100A8 is significantly upregulated in WT but not in *Hsf1* null colons at day 20 of the AOM-DSS protocol (Supplementary Fig. 7j, k).

FN1 and SERPINH1 were expressed in normal colons, yet their expression pattern changed dramatically in response to inflammation in WT, but not in *Hsf1* null colons (Supplementary Fig. 7l, m). In normal colons, FN1 was expressed mostly in the walls of the colon crypts and in the muscle, and SERPINH1 was expressed in goblet cells. Inflammation-induced massive expression of both of these proteins in the infiltrating stroma of WT mice but not *Hsf1* null mice (Supplementary Fig. 7l, m). Notably, the genes encoding for FN1, SERPINH1, and MMP9 were previously shown to be direct transcriptional targets of HSF1[20,37–39]. Taken together these findings suggest that inflammation induces the activation of HSF1 in stromal fibroblasts, leading to ECM remodeling through activation of ECM proteins, such as FN1, matrix remodeling enzymes such as MMP7 and MMP9, supporting chaperones such as the collagen chaperone SER-PINH1, and immune regulatory proteins such as LCN2 and S100A8. Some of these proteins are activated in a cell-autonomous manner (MMP7 and MMP9), while others are most likely activated by non-cell-autonomous signaling in neighboring and recruited immune cells. These events lead to remodeling of the ECM and, consequently, promote tumor progression.

**Loss of *Hsf1* impairs ECM deposition by colon fibroblasts, in vitro.** Our initial in vitro findings in MEFs (Fig. 1), together with our MxIF analysis of HSF1 localization and activation in the colon (Fig. 6), strongly indicate that, while HSF1 in cancer cells is important for tumor growth, and HSF1 in immune cells may contribute as well (Supplementary Fig. 4e–h), HSF1 in fibroblasts is a major player in driving the ECM rearrangements observed in

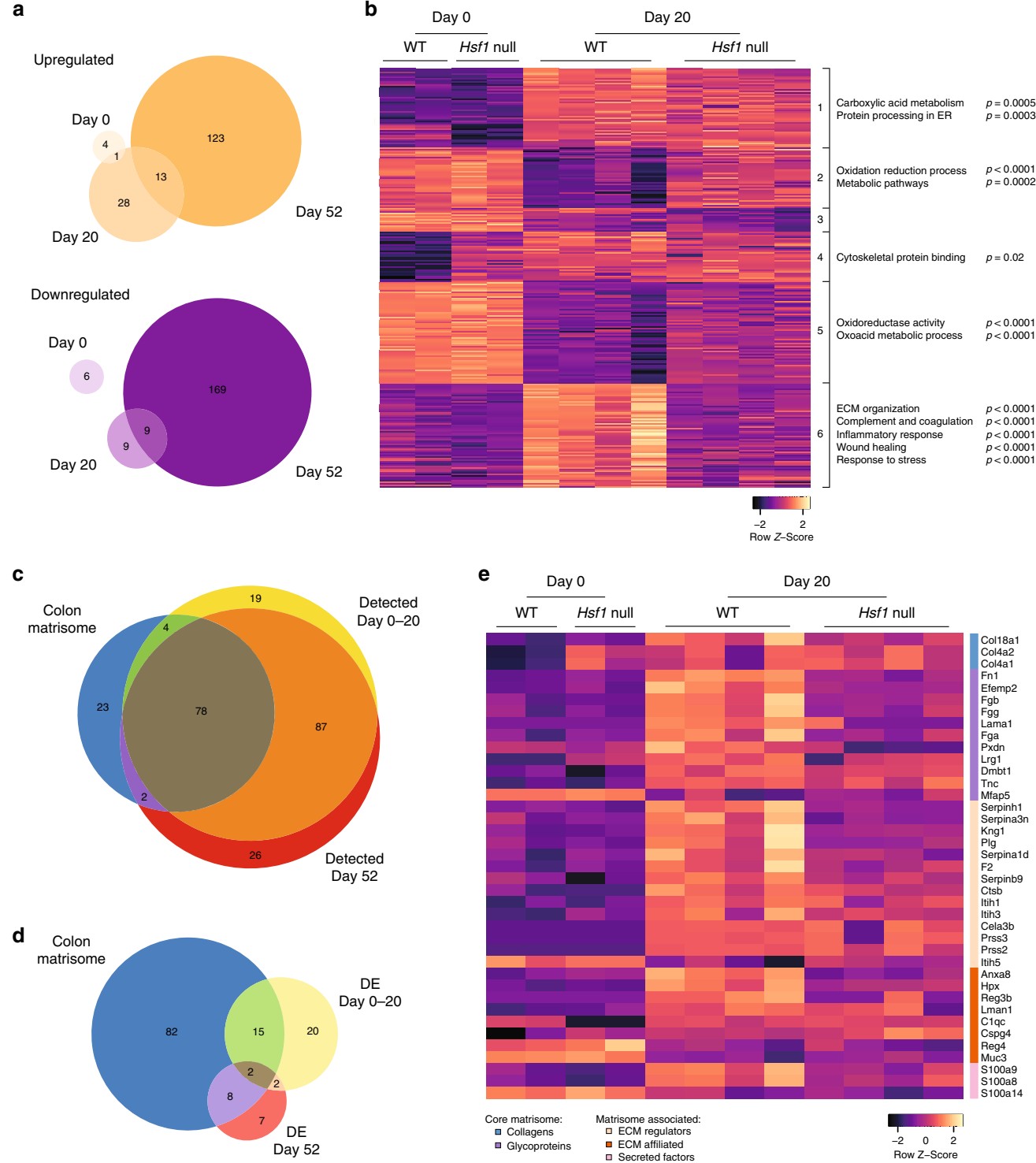

**Fig. 5 Mass spectrometry analysis reveals HSF1-dependent upregulation of proteins involved in inflammation and ECM remodeling in response to AOM-DSS.** Colons from non-treated (day 0) mice, or from mice treated with AOM-DSS for 20 or 52 days were analyzed by LC–MS/MS. **a** Venn-diagrams representing overlap between differentially expressed proteins upregulated (upper panel) and downregulated (lower panel) in WT vs. *Hsf1* null colons at different time points of the AOM-DSS treatment. **b** Heatmap of standardized mass spectrometry values of differentially expressed (DE) proteins (FC > 1.5; FDR < 0.1) between day 0 and day 20 WT and *Hsf1* null colons, clustered using *K*-means partition clustering into six clusters. Pathway analysis was performed using gProfiler and selected significant pathways are shown. See also Supplementary Data 3, 4. **c**, **d** Venn-diagrams representing overlap between the mouse colon matrisome and the matrisome proteins detected **c** and DE **d** in our mass spectrometry analysis. **e** Heatmap of standardized mass spectrometry values of DE matrisome proteins (FC > 1.5; FDR < 0.1) between day 0 and day 20 WT and *Hsf1* null colons, clustered by affiliation to core matrisome and matrisome-associated groups.

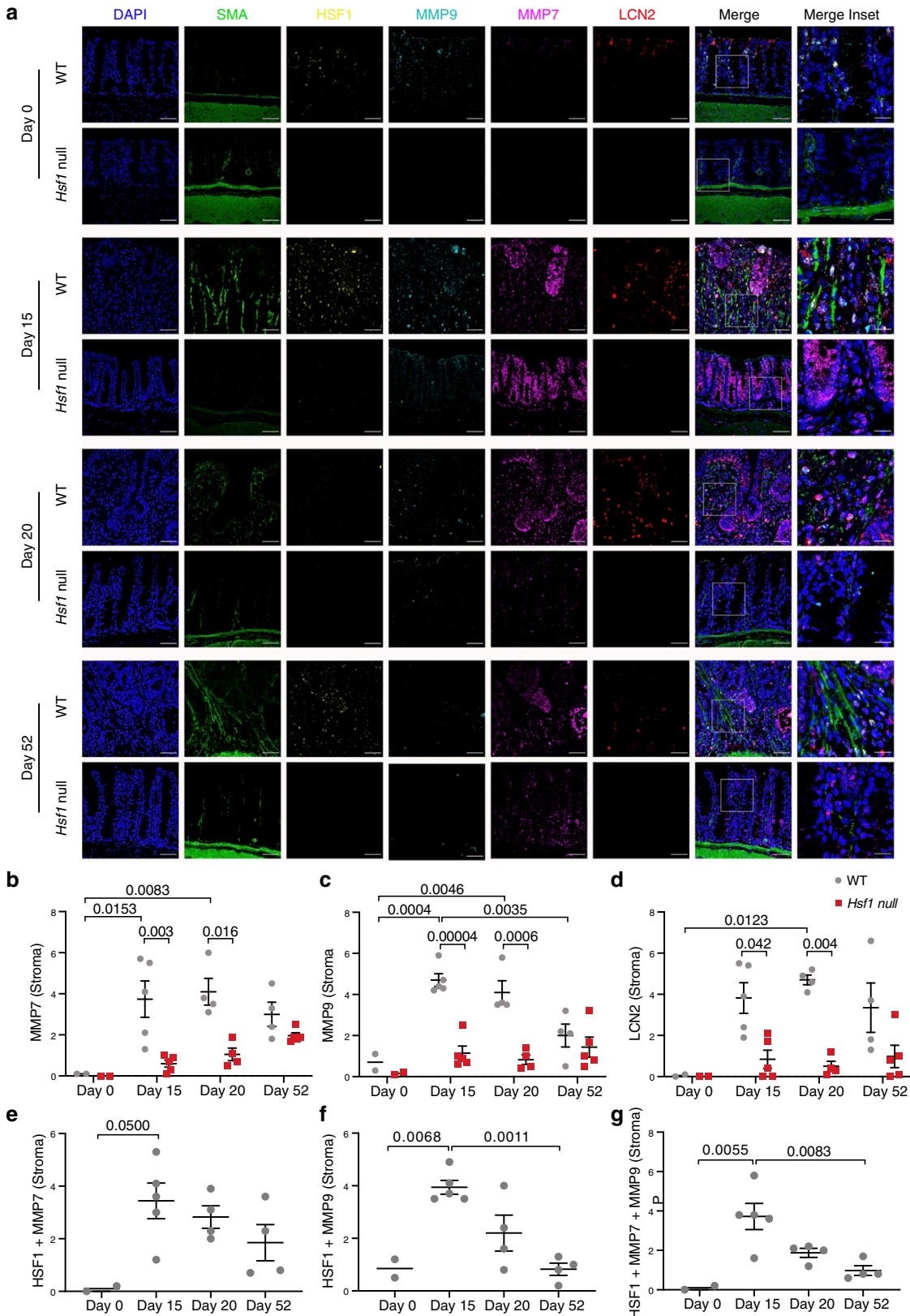

AOM-DSS-treated mice. To test this, we isolated colon fibroblasts from WT and *Hsf1* null mice at day 15 of the AOM-DSS protocol, allowed them to deposit fibrillar collagen, in vitro, in the presence of ARC38-conditioned media (see the "Methods" section) and measured collagen-deposition using SHG. Similar to MEFs, 15-day-treated WT colon fibroblasts deposited significantly more

collagen than fibroblasts from treated *Hsf1* null mice (Fig. 7a, b). We further confirmed these findings using a quantitative Sirius red staining protocol. In this assay, colon fibroblasts isolated from 15-day treated WT or *Hsf1* null mice were cultured for 5 days in the presence or absence of MC38-conditioned media, and the amount of fibrillar collagen produced by the cells was quantified

**Fig. 6 MMP7, MMP9, and LCN2 are upregulated in the stroma upon inflammation in an HSF1-dependent manner.** Colons from WT and *Hsf1* null mice were excised, fixed, and stained by multiplexed immunofluorescence (MxIF) using antibodies for the depicted proteins, at different time points along the AOM-DSS protocol. **a** Representative images are shown. Scale bar—50 μm. Inset scale bar—17 μm. **b–g** Stromal expression of each protein **b–d** and co-expression with stromal HSF1 **e–g** were scored in 5–7 representative images from each mouse, averaged, and are presented for each mouse in the group (see the "Methods" section for details). Results are presented as mean ± SEM, analyzed by two-way ANOVA and Bonferroni correction for multiple comparisons. For AOM-DSS-treated mice, n = 4 or 5 WT mice (day 52 and 20 or 15, respectively) and n = 5 or 4 *Hsf1* null mice (day 15 and 52, or day 20, respectively); n = 2 mice for control; results are presented for one of two independent experiments for day 0 and 15, and were combined from two independent experiments for days 20 and 52.

by Sirius red. Cells grown in control conditions (i.e. without growth factor-enriched MC38-conditioned media; see the "Methods" section) produced very little fibrillar collagen, regardless of the genotype. Upon induction with MC38-conditioned media, WT fibroblasts produced significantly more collagen than *Hsf1* null fibroblasts (Fig. 7c). WT fibroblasts also expressed and secreted more FN1, MMP7, and MMP9 than *Hsf1* null fibroblasts (Fig. 7d–f), supporting our in vivo observations (Fig. 6 and Supplementary Fig. 7). Cell-induced gel contraction by WT colon fibroblasts was similar to that of *Hsf1* null colon fibroblasts (Supplementary Fig. 8a), suggesting that HSF1 does not play an essential role in contractility. To demonstrate that ECM deposition by colon fibroblasts is HSF1-dependent, we used the synthetic small molecule CMLD011866 ((-)-aglaroxin C[40–42], Fig. 7g). This compound is a pyrimidinone variant of the rocaglate/flavagline natural product class, which also includes molecules such as rocaglamide A (RocA), a natural product translation-initiation inhibitor previously shown to inhibit HSF1's activity[43]. CMLD011866 itself has also been shown to inhibit the HSF1-dependent heat shock response with an $IC_{50}$ of 15.3 nM[40]. Treatment of WT colon fibroblasts with CMLD011866 did not affect their viability (Fig. 7h; lower panels). It did however significantly inhibit their ability to deposit ECM, in vitro, as measured by SHG (Fig. 7h, i), supporting the conclusion that HSF1 plays a key role in ECM deposition by fibroblasts.

Next we set to assess the role of HSF1-dependent matrix remodeling in supporting cancer cell proliferation. To that end we decellularized colons from WT and *Hsf1* null mice[44,45], so that only the matrix remained intact and all cells were lysed and removed (Supplementary Fig. 8b). We then added MC38 cancer cells and allowed them to re-cellularize the matrices for 40 h. SHG imaging of the ECM combined with IF imaging of the cells showed that MC38 cells re-cellularized 15-day AOM-DSS WT-derived colons much more efficiently than they did *Hsf1* null colons from 15-day treated mice (Fig. 7j, k). A higher number of cancer cells was found and they penetrated deeper into the matrix of WT colons versus *Hsf1* null colons (Fig. 7j; lower panels). Cancer cells actively remodeled the matrix as apparent by "holes" in matrix of 20-day AOM-DSS-treated WT mice but not in *Hsf1* null matrix from the same time point (Fig. 7l).

These in vitro studies support the hypothesis that AOM-DSS induces HSF1 activation in fibroblasts, leading to formation of a pro-tumorigenic ECM that promotes cancer growth.

**HSF1 is activated in human CAC.** To examine the relevance of our findings to human cancer, we performed immunohistochemical (IHC) staining of human CAC patient samples with antibodies for HSF1. To assess inflammation and fibrosis we also performed H&E, Masson's trichrome, and Sirius red staining of the same samples (Fig. 8a). Notably, all of these patients exhibited extensive fibrosis, as observed by Masson's trichrome and Sirius red staining. Indeed, we observed high nuclear HSF1 staining in both cancer cells and fibroblasts in these patients compared to control, suggesting that HSF1 is activated in human CAC

(Fig. 8a). Next, we asked whether the matrisome proteins highlighted by our mass-spec analysis may play a role in human colon cancer. To answer this question, we mined the human colon matrisome (see the "Methods" section and ref. [46]). We found that 127 out of 170 matrisome proteins detected in day 0–20, and 127 out of 179 matrisome proteins detected in day 52 were previously identified in the human colon matrisome (Fig. 8b and Supplementary Data 8). Furthermore, 70% of the matrisome proteins differentially expressed in our mass-spec analysis were identified in the human colon matrisome (Fig. 8c and Supplementary Data 8). Since the human colon matrisome contains data both from normal and from CRC tissue, we next asked whether proteins differentially expressed in our mouse model are upregulated in cancer. Indeed, the matrisome proteins differentially expressed in our mouse model were significantly enriched in proteins upregulated in human cancer (Fig. 8d, e). The expression of many of these proteins, including S100A8 and FN1, was higher in cancer compared to normal human colon, and 7 were not detected at all in the normal colon and were expressed only in CRC (Fig. 8d, e). We further compared our list of differentially expressed matrisome proteins to a recently published list of matrix proteins associated with disease severity in human ovarian cancer, and shown to be conserved in other human cancers[47]. 12 out of 60 proteins in the ovarian cancer list were differentially expressed in our matrisome list (Supplementary Data 10). Of these overlapping proteins, all of those positively correlated with disease severity were higher in WT vs. *Hsf1* null, and those negatively correlated with disease severity were higher in *Hsf1* null colons, supporting the conclusion that HSF1 positively regulates the expression of matrisome proteins contributing to ECM remodeling in cancer. Taken together these findings suggest not only that our model recapitulates early stages of colon cancer, but also highlight the potential role of stromal HSF1 in ECM remodeling in human CAC.

## Discussion
Long-term exposure to chronic inflammation can lead to fibrosis and cancer. The ECM plays a crucial role in this process, however the sequence and timing of events, and the molecular cues linking inflammatory signals with ECM remodeling and cancer are still not well elucidated. Here we find that the transcriptional master regulator HSF1 is activated during early stages of inflammation in the colon, and its activation leads to remodeling of the ECM, supporting the development of colon cancer. Using the AOM-DSS mouse model of CAC we analyzed ECM structure and composition at different time points along CAC initiation and progression in WT and *Hsf1* null mice, and found that both the structure and the composition of the ECM change at early, pre-malignant stages. These inflammation-induced changes were significantly inhibited by loss of HSF1, as was the consequent progression to CAC. Establishing the relevance of our experimental findings to human disease, we found high activation of stromal HSF1 in CAC patient samples, and high conservation of the HSF1-dependent proteomic ECM signature in human CRC.

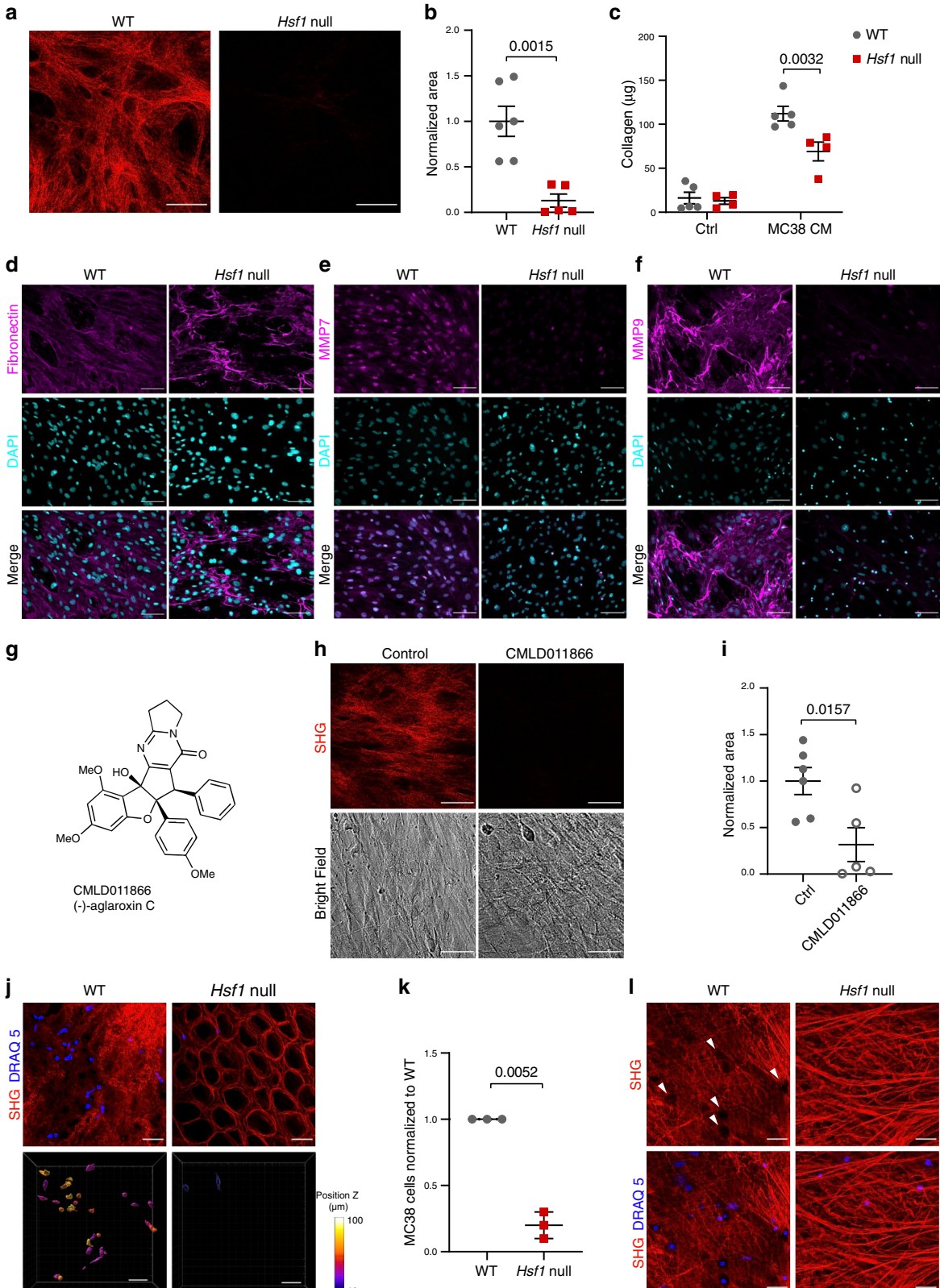

HSF1 has been previously linked to CAC, as its activation in cancer cells was shown to activate mTOR and increase glutaminolysis, thus promoting tumor growth[23]. Here we unravel a different, complementary role for HSF1, at the pre-malignant inflammatory stage, preparing the stromal niche. Our in vitro studies suggest that HSF1 is activated in fibroblasts, and that this activity is required for deposition of collagen fibers. Our mouse and human studies confirm that while HSF1 is activated in cancer cells, it is also activated in the stroma, mainly in fibroblasts but possibly also in other cell types in the TME. Future efforts,

**Fig. 7 Loss of *Hsf1* in colon fibroblasts impairs ECM secretion. a–f** Colons from day 15 AOM-DSS-treated WT or *Hsf1* null mice were excised, and fibroblasts were isolated and allowed to recover in culture for 5 days (see the "Methods" section). **a**, **b** Colon fibroblasts were induced to secrete ECM by 3–5 days incubation with MC38-conditioned media supplemented by growth factors and insulin. Representative SHG images are shown in **a**. The average area of collagen covered is quantified in **b** for $n = 6$ WT and $n = 5$ *Hsf1* null mice combined from three different experiments. Scale bar—50 μm. **c** Colon fibroblasts were cultured for 5 days in the presence or absence of MC38-conditioned media, and the amount of collagen secreted by the cells was quantified by Sirius red staining (see the "Methods" section). $n = 5$ WT and 4 *Hsf1* null mice combined from two different experiments. **d–f** Representative images of fibronectin **d**, MMP7 **e**, and MMP9 **f** and DAPI (nuclear) staining of colon fibroblasts cultured for 5 days in the presence of MC38-conditioned media. For **d–f** $n = 6$ WT and $n = 5$ **d**, **e** or 3 **f** *Hsf1* null mice, examined over two independent experiments. Scale bar—100 μm. **g** The molecular structure of (−)-aglaroxin C (CMLD011866). **h**, **i** WT colon fibroblasts were incubated with MC38-conditioned media supplemented by growth factors and insulin in the presence or absence of 3 nM CMLD011866 for 3–5 days, after which SHG imaging was performed. Representative images are shown in **h**. The average area of collagen covered is quantified in **i** for $n = 6$ WT mice and $n = 5$ *Hsf1* null mice, combined from three different experiments. Scale bar—50 μm. **j–l** Colons from day 15 **j–k** or day 20 **l** AOM-DSS-treated WT or *Hsf1* null mice were excised and decellularized. MC38 cancer cells were added and allowed to re-cellularize the matrices (see the "Methods" section) after which SHG imaging was performed. Scale bar—50 μm. **j, l** Representative SHG + IF images taken from the mucosal side for **j** (upper panels) and the muscularis externa side for **l**. SHG is shown in red, DRAQ5 nuclear staining is shown in blue. Heatmaps depicting the depth of invasion of cancer cells into the matrix are shown in **j** (bottom panels). **k** The average number of MC38 cells was calculated per mouse from 3 to 7 images averaged per area, each experiment was normalized to WT and the average of three biological replicates is presented. White arrows in **l** point to "holes" in the matrix where cancer cells have invaded. Results of **c** are presented as mean ± SEM, analyzed by two-way ANOVA and Bonferroni correction for multiple comparisons. For **b**, **i**, and **k**, Results presented as mean ± SEM, analyzed by an unpaired Student's *t*-test.

including development of highly specific Cre drivers for fibroblast and CAF subtypes[48], will allow us to directly assess the independent contribution of HSF1 in different stromal elements to inflammation and CAC. The temporal activation pattern of stromal HSF1 matches the course of disease, and the spatial activation pattern matches that of ECM remodeling proteins, suggesting that stromal HSF1 is the dominant player in the inflammation-driven process of ECM remodeling. The proteomic analysis was performed on bulk colons, including ECM, but also epithelial cells, fibroblasts, and immune cells. The finding that, within this bulk, ECM proteins are the most differentially upregulated proteins between WT and *Hsf1* null-treated colons further supports the conclusion that stromal HSF1 plays a key role in ECM remodeling in CAC.

*Hsf1* null mice are highly resistant to cancer in the AOM-DSS model. Are they also resistant to inflammation per se? Is inflammation induced in the *Hsf1* null mice? HSF1 has been attributed both pro-inflammatory and anti-inflammatory activities. For example, it can bind to the promoter of TNFα during proteotoxic stress and activate it[49], but was also recently suggested to inhibit TNFα through an alternative DNA-binding site[50]. Similarly, it binds to the promoter of IL-6, but could either activate its transcription (in response to heat-shock) or inhibit it (in response to LPS) depending on different binding partners[51,52]. It appears that different contexts (infection, heat-shock, or mutations) result in either pro-inflammatory or anti-inflammatory effects of HSF1. In the AOM–DSS model, different disease metrics including disease activity, colonoscopy, and histopathologic evaluation, as well as immune cell composition profiling indicate that *Hsf1* null mice are more moderately affected than WT by inflammation. Moreover, our proteomic analysis suggests that inflammation and wound healing are among the most differentially upregulated, HSF1-dependent pathways, suggesting that in the context of this model HSF1 promotes inflammation.

HSF1 is a transcription factor. Historically found to be activated by proteotoxic stress[53], it is now well known that HSF1 is also activated in other contexts and drives the transcription of genes other than chaperones[54,55]. We have previously shown that, in the context of cancer, HSF1 in fibroblasts drives the transcription of genes involved in inflammation, ECM remodeling, and wound healing[13]. Here we extend this analysis to unravel the proteomic consequences of stromal HSF1 activation, in cancer as well as in the preceding stage of inflammation. We propose that HSF1 is activated in response to inflammation, driving the transcription of genes encoding matrix proteins (FN1, LAMA1), matrix remodeling enzymes (MMP7, MMP9), and matrix chaperones (SERPINH1/HSP47), inducing inflammation in a non-cell-autonomous manner (S100A8/9, LCN2), and eventually leading to cancer.

Our comprehensive analysis of mouse colons at different time points along inflammation and tumor progression portrays dynamic temporal changes in ECM structure and composition, most of which are HSF1-dependent, and many of which precede the appearance of tumors. This dynamic shift in matrix protein expression is also reflected in the detection of multiple matrix proteins that are not part of the classic colon matrisome, and are only activated in the colon at the early stages of inflammation or later as inflammation progresses to cancer. Such is the case for S100A8/9, the calprotectin heterocomplex. This complex, a classic biomarker for disease activity in IBD[34], is specifically upregulated at the premalignant stage in an HSF1-dependent manner, and is not detected in the normal colon or in the malignant stage, neither in human nor in mice.

In humans, CAC develops in patients with IBD due to somatic mutations and the continuous exposure to inflammation, and it is very different, both in the spectrum of genetic alterations and in the etiology of the disease, from sporadic colon cancer[2,3]. These differences were revealed by several recent studies aimed at profiling the mutational landscape of CAC[2,3], yet we lack proteomic data to match these studies and uncover differences in protein expression between CAC and CRC. The vast majority of matrix proteins differentially expressed in our AOM-DSS model is conserved in the human colon matrisome[46] and many of the proteins are upregulated in sporadic CRC. Moreover, we find HSF1 activated in tumor-infiltrating stroma of CAC patients and not in normal colon fibroblasts. It is difficult to estimate the relative contribution of inflammation per se to the proteomic changes we observe. Recent proteomic analysis performed in mouse models of colitis, and confirmed in IBD patients, highlighted proteomic changes to the ECM that precede even the inflammatory symptoms[9]. CAC was not monitored in these mice or patients. Whether the changes we observe in the CAC model are associated with inflammation, cancer, or both, and which portion of these changes is specific to CAC and not to sporadic CRC remains to be determined and will be the subject of future studies. Nevertheless, our findings highlight HSF1 as a key mediator of the response to inflammation in the colon, and as such, an attractive potential therapeutic target in IBD and CAC.

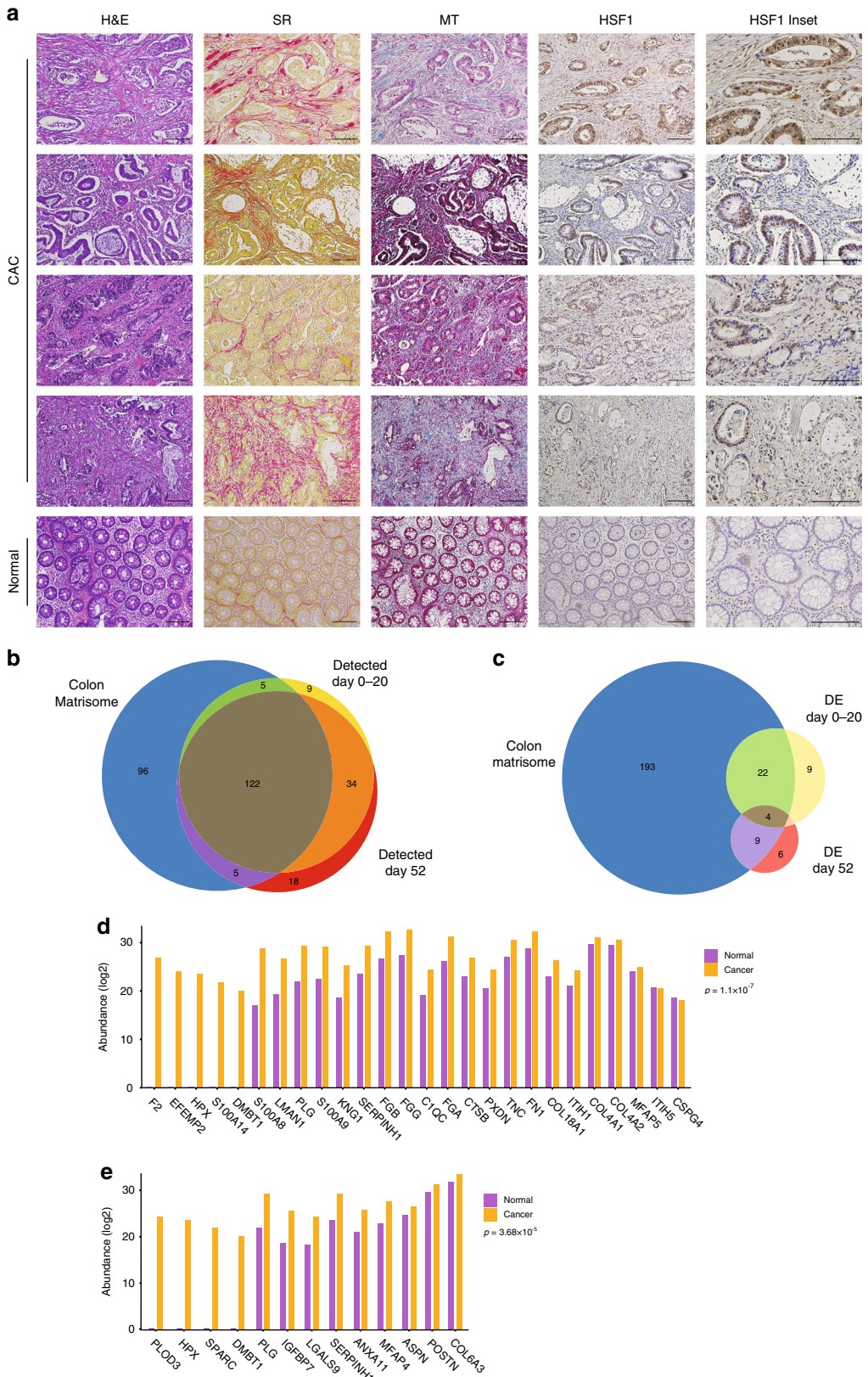

## Methods

**Ethics statement**. All clinical samples and data were collected following approval by the Sheba Medical Center Institutional Review Board (IRB, protocol # 6141-19-SMC and SMC-3305-16) and Ministry of Health (MOH) approval for the Israel National Biobank for Research (MIDGAM). All animal studies were conducted in accordance with the regulations formulated by the Institutional Animal Care and Use Committee (IACUC; protocol # 23590116-2, 34810317-2, 03960520-2).

**Human patient samples**. Whole tumor sections from four CAC patients and one normal adjacent control were retrieved and obtained from the Israel National Biobank for Research (MIDGAM; https://www.midgam.org.il/) under IRB #6141-19-SMC and SMC-3305-16. These samples were collected from patients who provided informed consent for collection, storage, distribution of samples, and data for use in future research studies.

**Fig. 8 HSF1 and its ECM-affiliated targets are expressed in human colon cancer. a** Formalin-fixed paraffin-embedded (FFPE) tumor sections from the colons of four colitis-associated cancer (CAC) patients were stained with H&E, Sirius red (SR), Masson's trichrome (MT), and antibodies for HSF1, and evaluated for inflammation and HSF1 staining in cancer and stromal regions by a pathologist. A normal human colon section was stained and evaluated as control. All sections were stained and imaged in parallel. Scale bar—150 μm. **b, c** Venn-diagrams representing overlap between the human colon matrisome and the matrisome proteins detected **b** and differentially expressed (DE; **c**) in our mass spectrometry analysis. **d, e** Peptide abundance in normal human colon vs. human colon cancer of matrisome proteins differentially expressed in our mass spectrometry analysis of day 0–20 **d** and day 52 **e**. Peptide abundance values were taken from refs. [46]. Statistical analysis was performed by paired *t*-test (two-sided).

**Mice**. *Hsf1* null mice and their WT littermates (BALB/c × 129SvEV, by Ivor J. Benjamin[56]), were maintained under specific-pathogen-free conditions at the Weizmann Institute's animal facility.

**Cell lines and primary cell cultures**. 4T1 mouse mammary carcinoma cells were kindly provided by Dr. Zvika Granot, Hebrew University of Jerusalem, and originally obtained from ATCC (ATCC® CRL-2539™). MC38 mouse colon cancer cells were kindly provided by Prof. Lea Eisenbach, Weizmann Institute of Science (originally from NCI). Primary MEFs derived from WT or *Hsf1* null mice (from the same colony), immortalized MEFs from the same strain, 4T1 cells and MC38 cells were cultured in Dulbecco's modified Eagle's medium (DMEM) (Biological industries, 01-052-1A) supplemented with 10% fetal bovine serum (FBS) and P/S. Primary fibroblasts from WT or *Hsf1* null colons were cultured in Roswell Park Memorial Institute 1640 (RPMI) (Biological industries, 01-100-1A) supplemented with 10% FBS, P/S, L-glutamine, Hepes (1 M), and sodium pyruvate solution (1 mM).

**Conditioned media**. Cancer cell conditioned medium was produced by plating 4T1 or MC38 cancer cells to confluence in 10 cm plates, 24 h later the medium was changed and collected 48 h later, centrifuged to remove cellular debris and stored at −20 °C until further use. Before use the conditioned medium was diluted 1:1 with complete medium.

**Induction of collagen deposition**. Immortalized or primary WT or *Hsf1* null MEFs were seeded at $1 \times 10^5$ cells/ml in 24-well plates on top of glass slides, primary fibroblasts from WT or *Hsf1* null colons were seeded on cell culture slides (CCS-8, MatTek). 48 h later (when the cells have formed a monolayer) the medium was replaced with collagen secretion medium[57]: L-ascorbic acid (cat. A8960, Sigma-Aldrich, Rehovot, Israel) 7.5 mg/ml, h-insulin (cat. 12585-0147.5, Gibco, NY, USA) 0.25 mg/ml, m-EGF (cat. PMG804, Gibco, NY, USA), 0.25 ng/ml mixed with conditioned medium (prepared as described above). For immortalized or primary MEFs the medium was changed every day, for 7 or 12 days, for primary colon fibroblasts, half of the medium was changed for 3–5 days, after which the glass slides were removed from the plate carefully and imaged using SHG for collagen imaging.

For HSF1 inhibition, CMLD011866 (aglaroxin C[40,41]) was first optimized to determine the maximal dose at which cell viability is not compromised (500 nM). Based on this optimization, 3 nM CMLD011866 was added daily to primary WT colon fibroblasts for 3–5 days and SHG imaging was performed.

**SHG imaging**. Distal portions of the colons or slides seeded with cells for ECM deposition assay were taken for SHG imaging using an upright Leica TCS SP8 MP microscope, equipped with external non-descanned detectors (NDD) HyD and acusto optical tunable filter (Leica microsystems CMS GmbH, Germany). Excitation—SHG signal was excited by a 885 nm laser line of a tunable femtosecond laser 680–1080 Coherent vision II (Coherent GmbH USA). Emission signal was collected using an external NDD HyD detector through a long pass filter of 440 nm. The transmitted signal was collected using a PMT detector in transmission position for general morphology. In addition, recellularization was imaged using excitation of HeNe 633 laser, with emission collection at 670–760 nm.

Images were acquired using the 8 kHz resonant scanner in a format of 1024 × 1024 (XY) through a HC PL APO 20X/0.75 CS2 objective, and the following parameters: scan speed—8000 Hz; zoom—1.3 for colons; and 3 for in vitro ECM deposition; line average—32; bit depth—8; FOV- X 0.416 μm, Y 0.416 μm; Z step - 0.684 μm; pixel size—416.25 nm (XY) for colons and 180.38 nm for in vitro ECM deposition. Z stacks were acquired using the galvo stage, with 0.68 μm intervals.

ECM deposition images from primary colon fibroblasts were acquired using a format of 1176 × 1176 (XY) with an HC FLUOTAR L 25X/0.95W VIS objective, and the following parameters: scan speed 400 Hz; Zoom 2; Line average—3; bit depth—8 FOV—X 0.188 μm, Y 0.188 μm; pixel size—188.45 nm.

Recellularization samples were acquired using a format of 2048 × 2048 (XY) with an HC FLUOTAR L 25X/0.95W VIS objective, and the following parameters: scan speed 700 Hz; zoom 1.3; line average—3; bit depth—16 FOV—X 0.166 μm, Y 0.166 μm; Z step—1 μm; pixel size—166.42 nm, Z stacks were acquired using the galvo stage, with 1 μm intervals.

The acquired images were visualized using LASX software (Leica Application Suite XLeica microsystems CMS GmbH).

**IF imaging**

*MxIF*. Samples from mouse colons were imaged with a LeicaSP8 confocal laser-scanning microscope (Leica Microsystems, Mannheim, Germany), equipped with a pulsed white-light and 405 nm lasers using a HC PL APO ×40/1.3 oil-immersion objective and HyD SP GaAsP detectors. 0.3 μm-thick optical sections were collected for each sample using the following fluorophores: DAPI (Ex. 405 nm Em. 424–457 nm); Opal 520 (Ex. 494 nm Em. 510–525 nm); Opal 570 (Ex. 568 nm Em. 575–585 nm); Opal 620 (Ex. 588 nm Em. 601–616 nm); Opal 650 (Ex. 638 nm Em. 647–664 nm); Opal 690 (Ex. 670 nm Em. 725–794 nm); and pinhole of 1 AU. Samples were acquired using a format of 2048 × 2048 (XY), and the following parameters: scan speed 600 Hz; zoom 1; bit depth—16; FOV—X 0.142 μm, Y 0.142 μm.

*IF*. Samples from primary colon fibroblasts were imaged with an inverted Leica DMI8 wide-field (Leica Microsystems, Mannheim, Germany), Leica DFC7000GT monochromatic camera, 20x/0.8 Air.

**DSS-induced colitis and AOM-DSS-induced cancer models**. DSS colitis was induced in 8-week-old male mice by 1.5% DSS (9011-18-1, MP Biomedicals, Santa Ana, OH) in the drinking water for 7 days followed by regular water. For AOM-DSS, mice were injected intraperitoneally with 10 mg/kg AOM (A5486, Sigma, Rehovot) followed by two cycles of 1.5% DSS in the drinking water, in days 7–12 and 28–33 (following Neufert et al. with modifications[25]). Disease progression was monitored by weight measurement, DAI, and colonoscopies either at day 20 (for 20-day-treated mice), or days 39 and 52 (for the full protocol). Sham-treated mice were injected with saline instead of AOM, and colonoscopies were performed as for the treated mice.

DAI was determined as previously published[58]. Colonoscopy was performed with a high-resolution mouse video endoscopic system (Carl Storz, Tuttlingen, Germany). Colitis colonoscopy score was determined using Murine endoscopic index of colitis severity (MEICS), which is based on five parameters: granularity of mucosal surface; vascular pattern; translucency of the colon mucosa; visible fibrin; and stool consistency[59]. At the endpoint of the experiments mice were sacrificed, colons were harvested and the colon weight and length was measured. Tumor burden was quantified post-mortem by macroscopic examination of the colons.

Each colon was cut longitudinally in two. The first half was cut again in half and the distal part was arranged from distal (Anus) to proximal, 0.3 cm were cut and the remaining portion was used for molecular biology methods in the following order: SHG, RNA, mass spectrometry (1, 1.5, 2.5 cm, respectively). The second longitudinal section was fixed in 4% paraformaldehyde over-night, and was used for histology methods. Paraffin-embedded sections were stained at the Weizmann histological unit using standard protocols for H&E, Masson trichrome and Sirius red. H&E slides were scored blindly by an expert veterinary pathologist (Dr. O. Brenner) following previously published parameters (percent of the involved tissue, crypt damage, layers that were affected, inflammation and regeneration)[58].

**FACS analysis of the innate immune profile**. WT and *Hsf1* null colons were washed through a strainer with PBS, dissociated by HBSS with 5 mM EDTA and 10 mM Hepes for 20 min, incubated for 10 min on ice with Ghost-Dye-Violet 450 viability dye (TONBO), washed and resuspended with MACS buffer (PBS (calcium and magnesium free), 0.05% BSA and 0.04% EDTA 0.5 M pH 8) for 10 min on ice with anti-mouse CD16/32 antibody. The cells were then stained with CD45-BV711, CD11b APC-cy7, Ly6G-PE-Dazzle 594, F4/80-FITC, and Ly6C–PerCP/Cy5.5 (see Supplementary Table 2 for details). FACS was performed using CytoFelx-S (Beckman Coulter) and analysis was performed with FlowJo 10.1.

**Chemical reagents**. ((-)-Aglaroxin C (CMLD011866) was synthesized according to published protocols[40,41].

**Mass spectrometry**. Mass spectrometry was carried out at the Smoler Proteomic Center at the Technion, Israel.

*Proteolysis*. Tissues for mass spectrometry were collected from the distal part of the colon after a brief PBS wash, snap-frozen in liquid nitrogen and stored in −80 °C until processing. Tissues were homogenized with an Omni-Th homogenizer in urea buffer containing: 8 M urea, 400 mM ammonium bicarbonate, and 10 mM DTT.

Homogenates were sonicated (5′, [10/10 on/off pulses], 90% energy. Sonics Vibra-Cell) and briefly centrifuged to pellet insoluble debris. Protein amount was estimated using Bradford readings. 20 µg protein from each sample were reduced with DTT (60 °C for 30 min), modified with 40 mM iodoacetamide in 100 mM ammonium bicarbonate (in the dark, RT, 30′) and digested in 2 M Urea, 100 mM ammonium bicarbonate with modified trypsin (V5111, Promega, WI, USA), overnight at 37 °C with a 1:50 enzyme-to-substrate ratio. An additional second digestion with trypsin was done for 4 h at 37 °C with a 1:100 enzyme-to-substrate ratio. The tryptic peptides were desalted using C18 tips (74–46, Harvard apparatus, MA, USA) dried and re-suspended in 0.1% formic acid. The peptides were resolved by reverse-phase chromatography on 0.075 × 300-mm fused silica capillaries (J&W scientific) packed with Reprosil reversed phase material (Dr. Maisch GmbH, Germany). The peptides were eluted with linear 180 min gradient of 5–28%, 15 min gradient of 28–95%, and 25 min at 95% acetonitrile with 0.1% formic acid in water at flow rates of 0.15 µl/min. Mass spectrometry was performed by Q Exactive plus mass spectrometer (Thermo Fischer Scientific) in a positive mode using repetitively full MS scan followed by high collision dissociation (HCD) of the 10 most dominant ions selected from the first MS scan.

*Analysis.* The mass spectrometry data from all the biological repeats was analyzed using the MaxQuant software 1.5.2.8[60] vs. the *Mus-Musculus* portion of the Uniprot database, with 1% FDR. The data was quantified by label-free analysis using the same software, based on extracted ion currents (XICs) of peptides enabling quantitation from each LC/MS run for each peptide identified in any of experiments. Statistical analysis of the identification and quantification of results was done using Perseus 1.5.2.4 software[61]. Day 52 samples ran separately from day 0 and day 20 samples, and were therefore also analyzed separately. LFQ intensities were filtered by minimum number of Razor + unique peptides ≤ 2. LFQ intensities were Log2 transformed and known contaminants were removed. Only the first majority protein ID was used for annotation, zero values were replaced with 15. Proteins for which there was no agreement of expression between the two samples of day 0 were removed from the analysis. For the day 0–20 data set, 3830 proteins passed filtration. Two-way ANOVA + multiple testing correction test (step-up) was applied to detect differentially expressed proteins between the *Hsf1* null and WT samples. A cutoff value of fold change >1.5 and FDR = < 0.1. was used and revealed 265 proteins as differentially expressed. For Day 52, 4234 proteins passed the filters above. Unpaired Students' *t*-test (two-sided) was applied to detect differentially expressed proteins, resulting in 313 proteins with FDR < 0.1.

Partitional clustering was applied on log2 transformed and standardized expression values, using the *k*-means algorithm (Euclidian method) for the day 0–20 data. Pathway analysis was done using the gProfiler web tool (e101_eg48_p14_baf17f0)[62].

*Matrisome analysis.* To create a list of all the matrisome proteins detected in our data we first compared our list of all detected proteins, and our lists of differentially expressed (DE) proteins, with comprehensive matrisome lists available through the Matrisome project for human and mouse[26]. Next we compared our lists of detected and DE matrisome proteins with the mouse colon matrisome[28], and with the human colon matrisome (compiled from normal colon, primary CRC and metastatic colon CRC[46]). Finally, we compared peptide abundance of our DE matrisome proteins in normal human CRC *vs* primary and metastatic CRC.

**Immunohistochemistry of human tissues.** Colons dissected as described above were fixed in 4% paraformaldehyde (PFA), processed and embedded in paraffin blocks, cut into 4–5 µm sections and immunostained as follows: formalin-fixed, paraffin-embedded (FFPE) sections were deparaffinized, treated with 0.3% $H_2O_2$ and antigen retrieval was performed by microwave (2 min full power, 1000 W, then 10 min at 30% of full power, and then cooling at RT for 10 min) with citrate acid buffer (pH 6.0). Slides were blocked with 10% normal horse serum, and anti-HSF1 antibodies were used (see Supplementary Table 2 for details). Visualization was achieved with 3,30-diaminobenzidine (DAB) as a chromogen (#SK4100, Vector Labs kit, CA, USA). Counterstaining was performed with Mayer-hematoxylin (MHS-16, Sigma-Aldrich, Rehovot, Israel). Images were taken with a Nikon Eclipse Ci microscope and Pannoramic Scan II scanner, ×20/0.8 objective (3DHISTECH, The Digital Pathology Company, Budapest, Hungary).

**Immunofluorescent staining of mouse and human tissues.** FFPE sections from mouse colons were deparaffinized, and incubated in 10% neutral buffered formalin (NBF prepared by 1:25 dilution of 37% formaldehyde solution in PBS) for 20 min in room temperature, washed (with PBS) and then antigen retrieval was performed by microwave (2 min full power, 1000 W, then 10 min at 30% of full power) with citrate buffer (pH 6.0). Slides were blocked with 10% BSA + 0.05% Tween20 and the antibodies listed in the table below were diluted in 2% BSA in 0.05% PBST and used in a multiplexed manner with the OPAL reagents, each one O.N. at 4 °C. The OPAL is a stepwise workflow that involves tyramide signal amplification. This enables simultaneous detection of multiple antigens on a single section by producing a fluorescent signal that allows multiplexed immunohistochemistry, imaging, and quantitation (Opal Reagent pack and amplification diluent, Akoya

Bioscience). Briefly, following over-night incubation with primary antibodies, slides were washed with 0.05% PBST, incubated with secondary antibodies conjugated to HRP for 10 min, washed again and incubated with OPAL reagents for 10 min. Slides were then washed and microwaved (as describe above), washed, stained with the next primary antibody or with DAPI in the end of the cycle and mounted. We used the following staining sequences: (1) αSMA, HSF1, MMP9, LCN2, MMP7; (2) SERPINH1, FN1. S100A8 was stained separately. Each antibody (see Supplementary Table 2 for details) was validated and optimized separately, and then multiplexed immunofluorescence (MxIF) was optimized to confirm that signals were not lost or changed due to the multistep protocol. The slides of sequence (1) were imaged with a DMi8 Leica confocal laser-scanning microscope, using a HC PL APO ×40/1.3 oil-immersion objective HyD SP GaAsP detectors. The slides of sequence (2) and of S100A8 were imaged with the Pannoramic Scan II scanner, ×20/0.8 objective (3DHISTECH, The Digital Pathology Company, Budapest, Hungary).

**Isolation of primary colon fibroblasts.** Primary colon fibroblasts were produced from WT or *Hsf1* null mice on day 15 of the AOM-DSS protocol using a standard protocol[63]. Briefly, mice were euthanized and their colons were removed and washed with cold PBS. The colons were opened and epithelial cells were denuded by three repeated washes of 15 min at 37 °C, 250 rpm in a tilted shaker-incubator, in HBSS (Cat. H6648, Sigma-Aldrich, Rehovot, Israel) with 5 mM EDTA and P/S, after which the tissue was washed again with cold PBS and put in complete RPMI 1640 medium (described above) with 3.5 mg/ml Collagenase D (Cat. 11088866001, Merck, Rehovot, Israel) and 10 U of Dispase II (Cat.D4693, Sigma-Aldrich, Rehovot, Israel) for 1 h in 37 °C at 250 rpm in a flat shaker-incubator. Tissues were then spun down at 300×g for 5 min and reconstituted with 5 ml complete RPMI 1640 and seeded on 60 mm plates coated with collagen I (Cat. C3867, Sigma-Aldrich, Rehovot, Israel). Cells were incubated for 5 days before seeding for experiments.

**Total collagen detection in vitro.** WT and *Hsf1* null primary colon fibroblasts were seeded at $2 \times 10^5$ cells/ml in 96-well plates. 24 h later (when the cells have formed a monolayer) the medium was replaced with either the same medium (as control) or with MC38 condition medium (prepared as described above) supplemented with collagen secretion medium[57] (L-ascorbic acid (cat. A8960, Sigma-Aldrich, Rehovot, Israel) 7.5 mg/ml; h-insulin (cat. 12585-0147.5, Gibco, NY, USA) 0.25 mg/ml; m-EGF (cat. PMG804, Gibco, NY, USA) 0.25 ng/ml. Half of the medium was changed every day, for 5 days, after which cell lysates were removed with acetic acid 0.5 M, and Sirius Red total collagen detection assay kit (cat. 9062, Chondrex, WA, USA) was used according to manufactures' protocol to measure fibrillar collagen content.

**Immunofluorescent staining of primary colon fibroblasts.** Primary colon fibroblast were seeded at $1 \times 10^5$ cells/ml in 24-well plates on top of glass slides. 48 h later (when the cells have formed a monolayer) the medium was replaced with collagen secretion medium as described above, for 5 days, after which the cells were fixed with 4% paraformaldehyde (PFA) for 10 min, washed and permeabilized using 0.1% Triton X-100 for 10 min. Samples were blocked with 1% BSA in 0.05% PBST for 1 h, incubated with primary antibodies (FN1, MMP7, MMP9) overnight, and with AF647 secondary antibodies for 1 h. Samples were imaged as described above.

**Decellularization and recellularization of colon tissue.** Colon tissue decellularization was performed according to a protocol adapted from Silva et al.[45]. Briefly, ~10 mm × 5 mm of colon tissue with agitation at 165 rpm (shaker-incubator with orbit diameter of 20 mm) and at 25 °C, unless specified otherwise. Following thawing, colon tissue was incubated for 18 h in a hypotonic buffer (10 mM Tris HCl/0.1% EDTA, pH 7.8) and washed three times in PBS (1 h per wash). Samples were then immersed in detergent solution (0.2% sodium dodecyl sulfate (SDS)/10 mM Tris–HCl, pH 7.8) for 24 h, washed three times (20 min per wash) with the hypotonic wash buffer (10 mM Tris–HCl, pH 7.8) and incubated for 3 h, at 37 °C, in DNAse solution (50 U/mL DNAse I (Cat. LS00200, Worthington, NJ, USA)/10 mM Tris–HCl, pH 7.8). Three final washes in PBS were performed (20 min per wash) to remove residual detergent and DNAse. Decellularized matrices were kept under sterile conditions, at 4 °C in PBS supplemented with 1% P/S until use. Recellulriztion was performed as following: The decellularized tissue was incubated in DMEM for 1 h at 37 °C, after which it was gently deployed at a fixed orientation (under microscopy guidance) on a Petri dish (60 mm) and $0.5 \times 10^6$ MC38 cells were seeded through a strainer (40 µm) in 1 ml of medium. After 40 or 72 h for day 15 or 20, respectively, the colon tissues were fixed with 4% PFA for 10 min, washed gently with PBSx1, stained with DRAQ5 for 10 min and washed again. The samples were imaged as described above.

**RNA sequencing.** Tissues for RNA-seq were collected from the distal part of the colon after a brief PBS wash, snap-52frozen in liquid nitrogen and stored in −80 °C until processing. Tissues were homogenized using Fisherbrand bead mill and RNA was extracted using RNeasy fibrous tissue mini kit according to manufactures protocol (Cat. 74704, Qiagen, MD, USA). RNA libraries were prepared using the SENSE mRNA-seq library prep kit (Cat. 001, Lexogen, Vienna, Austria). Libraries

were sequenced using the NovaSeq 6000 machine (Illumina, CA, USA) and reads were aligned to the mouse reference genome (mm10) using STAR v2.4.2a[64] with default parameters. Counts were normalized using Deseq2[65]. RNA sequencing results are detailed in Supplementary Data 5. Hierarchical clustering was performed using Euclidian distance on differentially expressed genes which were filtered with the following parameters: baseMean >5, padj < 0.05 and |logfoldchange| >1. Analysis was performed by R software v.3.6. Pathway analysis was performed using the g-profiler web tool[62]. Significant pathways were determined if $p < 0.05$ (for details see Supplementary Data 7). Comparison of the RNA-seq data to ECM proteins was done using the mouse matrisome data base[26] (for more information see Supplementary Data 9).

### Image analysis

*MxIF*. MxIF images from the distal half of the colon were analyzed using Fiji image processing platform[66]. Analysis of MxIF staining for HSF1, SMA, LCN2, MMP7, MMP9 was performed as following: five slices—were Z projected (Average) and linear spectral unmixing was performed. Manual scoring was then performed in a blinded manner by three independent experts for epithelial cells and stromal cells separately (excluding the submucosa and tunica muscularis), considering the number of positive cells, co-expression between markers, and the total amount of epithelial cells or stromal cells in the image. These scores were averaged and are presented. For S100A8, FN1, and SERPINH1 staining regions of interest (ROIs) were manually depicted using QuPath[67] (0.2.0-m8) to include all intact tissue areas in the vertical orientation and exclude regions of tunica muscularis (due to its tendency to display nonspecific staining). Following background subtraction using a rolling ball all the channels were automatically thresholded (Triangle method). For S100A8 quantification, Threholded areas were normalized to ROI. For FN1 and SERPINH1 changes in localization were evaluated manually and independently by two blinded experts and representative images were selected accordingly.

### SHG

*Collagen covered area*. Collagen covered area was analyzed using LAS X 3D Analysis, starting with the following pre-filter processing: (1) enhance white detail (size 2); (2) stretch the histogram; and (3) noise removal median (size 1). Next, we used a fixed threshold, which was set manually, to detect the collagen matrix.

**Spatial analysis of distances between crypts**. We started with Max projection of the 3D stack, and trained Ilastik (1.3.3b2)[68] AutoContext Pixel Classifier to classify pixels into crypts vs. collagen fibers, on selected images. A dedicated Fiji macro[66] applied this classifier to each image, and segmented the individual crypts using hysteresis thresholding from 3D ImageJ Suite plugin[69]. Holes in the crypt objects were filled and objects smaller than 300 μm² were discarded. For each image we measured the number of crypts completely included in the image, and the average size of those crypts.

To quantify fibrosis we calculated the border-to-border shortest distances between neighboring crypts using 3D image-J suite plugin[69]. We further looked at the following measures: the border-to-border distance to the nearest crypt, and the number of neighboring crypts within 20 μm from the crypt border. Images of segmented crypts color-coded by these measurements were created using MorpholibJ plugin[70].

The values for crypts on the border of the image were not used in the statistics, but those crypts were taken into account as neighbors of in-image crypts. This automatic process segmented correctly most of the crypts. Further manual correction was done to fix for segmentation errors. All the measurements were recalculated based on the corrected segments. The macro allows for this by saving crypts contours in a file, and by providing an update mode to calculate the measurements from a segments file. The macro is available on Github [https://github.com/WIS-MICC-CellObservatory/Crypts_SpatialOrganization/].

**Recellularization of MC38 cells on WT or *Hsf1* null colons**. The number of MC38 cells growing on WT or *Hsf1* null colons was calculated using Imaris (9.6.0, Bitplane) "surface" module to segment the nuclei. Segmentation was done using absolute intensities of the DAPI channel, default smoothing settings, an automatic threshold, and splitting touching objects according to a seed point diameter of 7 μm. A fixed quality value for selecting seed points was selected manually and applied for all the images. Small objects were discarded using a fixed threshold. Color-coded maps of depth were created by coloring each nucleus by its z-position value.

It should be noted that WT colon samples are more fibrotic, and are therefore thicker than *Hsf1* null samples. Figure 7j top panel shows a merged image of SHG channel (averaged section of 5 μm) and nuclei channel (averaged across all section).

**Collagen contraction assay**. For the gel contraction assay $0.5 \times 10^6$ primary colon fibroblast were seeded in 100 μl volume of Matrigel solution containing 0.1% serum-FBS, 0.2% 5x RPMI buffer (52 mg RPMI, 20 mg NaCHO₃, 10 μl HEPES 1 M, and 0.9 ml DDW), 0.2% Matrigel (Corning, #354230), 0.4% Collagen type I (Corning, #354230), and 0.1% PBS. The mixed collagen-cells were plated in 96-well plates and incubated for 15 min at 37 °C after which 'control medium' (RPMI

1640) or 0.75% 'conditioned medium' (as described above) mixed with regular medium was added. Plates were scanned 3 days later and the area that covered with gel was measured.

**Statistical analysis**. Statistical analysis and visualization were performed using R (Version 3.6.0, R Foundation for Statistical Computing Vienna, Austria) and Prism 8.2.0. Statistical significance tests were performed as described in each figure legend. Unless stated otherwise all ANOVA tests were significant with $p$ value < 0.05. $p$ values > 0.05 were defined as not significant and are not presented in the graphs. We excluded 1 WT mouse from crypt size (11,000 μm²) quantification presented in Fig. 3, since it was an outlier (statistically significantly different from group average).

## Data availability

The proteomics data generated and analyzed in this study has been deposited in the Proteomics Identification Database (PRIDE) number PXD022207. RNA-sequencing data has been deposited in Gene Expression Omnibus (GEO) number GSE158276. The remaining data are available within the Article, Supplementary Information or available from the authors upon request.

## Code availability

Script and auxiliary data needed to reconstruct "spatial analysis of distances between crypts" analysis files are available from GitHub [https://github.com/WIS-MICC-CellObservatory/Crypts_SpatialOrganization/] and from Zenodo [https://doi.org/10.5281/zenodo.4172577].

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

## Acknowledgements

Mass spectrometry was carried out at the Smoler Proteomic Center at the Technion, Israel. Pathological evaluation was carried out by Dr. Ori Brenner, Veterinary resources, WIS. We thank Dr. Yoseph Addadi at the MICC cell observatory, WIS and Vladimir Kiss at the Department of Biomolecular Sciences, WIS, for their assistance with imaging. We thank Dr. Z. Granot (HUJI) and Prof. L. Eisenbach (WIS) for cell-lines. We thank Prof. I. Sagi (WIS) and members of the Scherz-Shouval lab for valuable input on the manuscript. The SHG and IF images were acquired at the Advanced Optical Imaging Unit, the de Picciotto-Lesser Cell Observatory in memory of Wolfgang and Ruth Lesser unit at the Moross Integrated Cancer Center Life Science Core Facilities, Weizmann Institute of Science. Work at Boston University is supported by NIH grants R35GM118173 and U01TR002625. CAC work in MSKCC is supported by the Crohn's and Colitis Foundation; Liane Ginsberg Foundation; STARR Cancer Consortium Grant, and NCI Core Grant to MSKCC P30 CA008748. E.S. and E.W. are supported by the Francis Crick Institute, which receives its core funding from Cancer Research UK (FC010144), the UK

Medical Research Council (FC0010144), and the Wellcome Trust (FC010144). R.S.S. is supported by the Israel Science Foundation (grant nos. 401/17 and 1384/1), the European Research Council (ERC grant agreement 754320), the Minerva foundation, the Israel cancer research fund, the Laura Gurwin Flug Family Fund, the Peter and Patricia Gruber Awards, the Comisaroff Family Trust, the Estate of Annice Anzelewitz, and the Estate of Mordecai M. Roshwal. R.S.S. is the incumbent of the Ernst and Kaethe Ascher Career Development Chair in Life Sciences.

## Author contributions

O.L.-G. designed, performed, and analyzed experiments, and wrote the manuscript. H.L., designed, performed and analyzed experiments, and wrote the manuscript. R.W.-D., M.P.-F., S.M., and G.F. designed, performed, and analyzed experiments. E.W. and E.S. designed image analysis and provided intellectual input. Y.S. designed and performed statistical and image analysis. L.B., W.Z., and J.A.P. provided reagents and intellectual input. O.G. designed and performed image analysis. R.N. assisted with image acquisition and designed image analysis. L.K., R.Y., I.L, D.S.S, and D.K. provided clinical samples and intellectual input. R.S.-S. directed the study, designed and analyzed experiments, and wrote the manuscript.

## Competing interests

The authors declare no competing interests.
