## [Peer Review File · Nature Communications]

REVIEWERS' COMMENTS

Reviewer #1 (Remarks to the Author):

Authors did a very good job on the revision of the previous version. New experiments and analyses are done, together with text modifications. The very fibroblast-specific role of HSF1 remains a "weaker" point of the paper, but authors are correct that there is not much can be done additionally about it within the reasonable revision timelines ((a new (or several new) Cre lines and Hsf1 flox mice) to unequivocally address the role of HSF1 specifically in colonic fibroblasts. I would be interested to know other reviewer's opinion but I would recommend acceptance of the paper

Reviewer #2 (Remarks to the Author):

The authors should be commended for their efforts to address many of the outstanding critiques raised in the first round of reviews. For the most part they have addressed this reviewer's concerns. I request that the data showing no impact on fibroblast contractility be included in supplemental figure data when the article is published since this is an important finding. Similarly, CAF marker comparisons should also be documented. I am disappointed the authors made no effort to conduct the suggested tissue specific manipulations - and their contention that col1Cre is not able to efficiently delete transgenes in stromal fibroblasts - is simply incorrect - we have personal experience working with this model and can attest to its suitability. That being said I am satisfied that the manuscript is now suitable for publication. I do request that the language be toned down regarding the authors' conclusions regarding stromal HSF1.

Reviewer #3 (Remarks to the Author):

Dear authors,

Thank you for addressing most of my comments, and also for taking into consideration the comments from the other referees. As a second round of review, I evaluated the manuscript in the context of my previous comments. I note, and understand that providing the evidence of the role of HSF1 in CAC, using a specific Cre/lox mice model in fibroblast, is a challenge, and would take a considerable amount of time accordingly to a review process.

The revised manuscript is largely improved when compared to the first draft, and I have no further concern to be addressed.

Response to reviewer comments

Below we include a point-by-point response to the reviewers. In particular, we addressed the request to discuss the limitations associated with the lack of a fibroblast specific conditional KO mouse model and we included the additional data suggested by Reviewer #2 in the manuscript.

Reviewer #1 (Remarks to the Author):

Authors did a very good job on the revision of the previous version. New experiments and analyses are done, together with text modifications. The very fibroblast-specific role of HSF1 remains a "weaker" point of the paper, but authors are correct that there is not much can be done additionally about it within the reasonable revision timelines ((a new (or several new) Cre lines and *Hsf1* flox mice) to unequivocally address the role of HSF1 specifically in colonic fibroblasts. I would be interested to know other reviewer's opinion but I would recommend acceptance of the paper

Response: We thank the reviewer for this positive assessment of our revised manuscript.

Reviewer #2 (Remarks to the Author):

The authors should be commended for their efforts to address many of the outstanding critiques raised in the first round of reviews. For the most part they have addressed this reviewer's concerns. I request that the data showing no impact on fibroblast contractility be included in supplemental figure data when the article is published since this is important findings. Similarly, CAF marker comparisons should also be documented. I am disappointed the authors made no effort to conduct the suggested tissue specific manipulations - and their contention that *col1Cre* is not able to efficiently delete transgenes in stromal fibroblasts - is simply incorrect - we have personal experience working with this model and can attest to its suitability. That being said I am satisfied that the manuscript is now suitable for publication. I do request that the language be toned down regarding the authors conclusions regarding stromal HSF1.

Response: We thank the reviewer for commending us, and are happy to learn that most concerns have been addressed.

1. We included the data showing no impact on contractility to revised Supplementary Figure 8, and included the following text in the revised manuscript (rows 764-767):
“Cell-induced gel contraction by WT colon fibroblasts was similar to that of *Hsf1* null colon fibroblasts (Supplementary Figure 8a), suggesting that HSF1 does not play an essential role in contractility.”
2. To address the limitations associated with the lack of a fibroblast-specific conditional KO mouse model we added the following sentence to the discussion (rows 1101-

1104): “Future efforts, including development of highly specific Cre drivers for fibroblast and CAF subtypes⁴⁸, will allow us to directly assess the independent contribution of HSF1 in different stromal elements to inflammation and CAC.” We have also toned down our conclusion regarding HSF1’s fibroblast-specific activation by modifying the abstract. Specifically, we omitted the sentence referring to activation of HSF1 in stromal fibroblasts in the gut.

Reviewer #3 (Remarks to the Author):

Dear authors,

Thank you for addressing most of the my comments, and also for taking into consideration the comments from the other referees. As a second round of review, I evaluated the manuscript in the context of my previous comments. I note, and understand that providing the evidence of the role of HSF1 in CAC, using a specific Cre/lox mice model in fibroblast, is a challenge, and would take a considerable amount of time accordingly to a review process.

The revised manuscript is largely improved when compared to the first draft, and I have no further concern to be addressed.

Response: We thank the reviewer for this positive assessment of our revised manuscript, and are happy to learn that no further concerns remain to be addressed.